# Zebrafish fin regeneration involves generic and regeneration-specific osteoblast injury responses

Ivonne Sehring[1]*, Hossein Falah Mohammadi[1], Melanie Haffner-Luntzer[2], Anita Ignatius[2], Markus Huber-Lang[3], Gilbert Weidinger[1]*

[1]Institute of Biochemistry and Molecular Biology, University of Ulm, Ulm, Germany; [2]Institute of Orthopaedic Research and Biomechanics, University Hospital Ulm, Ulm, Germany; [3]Institute of Clinical and Experimental Trauma-Immunology (ITI), University Hospital Ulm, Ulm, Germany

**Abstract** Successful regeneration requires the coordinated execution of multiple cellular responses to injury. In amputated zebrafish fins, mature osteoblasts dedifferentiate, migrate towards the injury, and form proliferative osteogenic blastema cells. We show that osteoblast migration is preceded by cell elongation and alignment along the proximodistal axis, which require actomyosin, but not microtubule (MT) turnover. Surprisingly, osteoblast dedifferentiation and migration can be uncoupled. Using pharmacological and genetic interventions, we found that NF-κB and retinoic acid signalling regulate dedifferentiation without affecting migration, while the complement system and actomyosin dynamics affect migration but not dedifferentiation. Furthermore, by removing bone at two locations within a fin ray, we established an injury model containing two injury sites. We found that osteoblasts dedifferentiate at and migrate towards both sites, while accumulation of osteogenic progenitor cells and regenerative bone formation only occur at the distal-facing injury. Together, these data indicate that osteoblast dedifferentiation and migration represent generic injury responses that are differentially regulated and can occur independently of each other and of regenerative growth. We conclude that successful fin bone regeneration appears to involve the coordinated execution of generic and regeneration-specific responses of osteoblasts to injury.

*For correspondence:
Ivonne.sehring@uni-ulm.de (IS);
gilbert.weidinger@uni-ulm.de (GW)

**Competing interest:** The authors declare that no competing interests exist.

## Editor's evaluation

This work is of interest to readers in the field of bone regeneration, and more broadly to readers in the field of tissue repair and regenerative medicine. The authors took advantage of a well–established in vivo model, live imaging, pharmacological inhibition, and genetic strategies to dissect the interrelations of key cellular events in zebrafish fin regeneration. The finding of how distinct generic injury responses are differentially regulated and are functioning independently from each other, is a valuable piece of information for the community.

## Introduction

For humans and other mammals, the traumatic loss of a limb represents an irreversible and permanent defect, as lost appendages cannot be restored. In contrast, teleost fish and urodele amphibians are able to fully regenerate limbs/fins. Thus, the zebrafish caudal fin has become a popular model to study the restoration of bone tissue (*Gemberling et al., 2013*; *Pfefferli and Jaźwińska, 2015*). The fin skeleton consists of dermal (directly ossifying) skeletal elements, the fin rays or lepidotrichia, which make up the largest part of the fin, and endochondral parts close to the body. Rays are segmented

with flexible joints, and each segment consists of two concave hemirays which are lined by a single layer of osteoblasts on the inner and outer surface (*Figure 1A*). Within 1 day after fin amputation, a wound epidermis covers the injured tissue. Next, a blastema forms atop of each ray, which contains the proliferative source cells for regeneration of the ray skeleton. Within hours after amputation, osteoblasts close to the amputation plane dedifferentiate, that is, they downregulate the expression of the mature osteoblast marker *bglap*, upregulate preosteoblast markers like *runx2*, and become proliferative (*Knopf et al., 2011*; *Sousa et al., 2011*; *Stewart and Stankunas, 2012*). Furthermore, osteoblasts relocate towards the amputation plane and beyond to contribute to the blastema (*Knopf et al., 2011*; *Geurtzen et al., 2014*). In the regenerate, these dedifferentiated osteoblasts remain lineage-restricted and redifferentiate to osteoblasts (*Knopf et al., 2011*; *Stewart and Stankunas, 2012*). While both osteoblast dedifferentiation and migration occur in response to fin amputation and bone fractures in zebrafish (*Knopf et al., 2011*; *Geurtzen et al., 2014*), the interrelation of these processes is not understood; specifically, whether dedifferentiation is a requirement for migration is not known.

Cell migration requires formation and retraction of membrane protrusions, which are regulated by the dynamic actomyosin network, while microtubuli play a role in organising the polarisation of migrating cells (*Petrie et al., 2009*). Bone tissue is permanently turned over by bone remodelling, a process of alternating bone resorption and bone formation (*Kular et al., 2012*). For bone formation, osteoblast precursors migrate to the resorbed sites (*Dirckx et al., 2013*). Similarly, during mammalian fracture healing, osteoblasts are recruited to the site of injury (*Thiel et al., 2018*). Several factors have been shown to act as chemoattractants for osteoblasts in vitro, but few have been confirmed to play a role in vivo (*Dirckx et al., 2013*; *Thiel et al., 2018*). One candidate osteoblast guidance cue is the complement system, which represents the major fluid phase part of innate immunity and is activated immediately after injury. It consists of more than 50 proteins, including serial proteases, whose activation leads to the formation of the peptides C3a and C5a, generated from the precursors C3 and C5, respectively (*Thorgersen et al., 2019*). While the majority of C3 and C5 precursors are expressed in the liver and distributed to the periphery via the circulation (*Merle et al., 2015*), C3 and C5 mRNA are also expressed by human osteoblasts, which in addition can cleave native C5 into C5a (*Ignatius et al., 2011b*). During bone fracture healing in mammals, the receptor for C5a (C5aR) is expressed by osteoblasts, and C5a can act as chemoattractant for osteoblasts in vitro (*Ignatius et al., 2011b*). In C5-deficient mice, bone repair after fracture is severely impaired (*Ehrnthaller et al., 2013*); however, whether this is due to defects in osteoblast migration is not yet known.

In this study, we analysed injury-induced osteoblast migration in vivo in the regenerating zebrafish fin. We show that osteoblast cell shape changes and migration depend on a dynamic actomyosin cytoskeleton, but not on microtubuli turnover. Pharmacological interference with C3a and C5a suggests that the complement system regulates osteoblast migration in vivo. Using genetic and pharmacological manipulation of NF-κB, retinoic acid (RA), and complement signalling, we found that dedifferentiation and migration can be uncoupled and are independently regulated, suggesting that dedifferentiation is not a prerequisite for migration. Furthermore, we established a novel injury model in which an internal bone defect within fin rays allows us to study osteoblast behaviours at proximally and distally facing injuries. Intriguingly, osteoblast migration and dedifferentiation occur at both injury sites, yet only at the distal injury a preosteoblast population forms and only here regenerative growth commences. We conclude that osteoblast migration and dedifferentiation represent generic injury responses in zebrafish, and that successful bone regeneration depends on additional, regeneration-specific events.

## Results

### Osteoblasts elongate, align along the proximodistal axis, and migrate towards the amputation plane

In non-injured caudal fins, differentiated osteoblasts expressing the markers *bglap* and *entpd5* line the two segmented hemirays that together form the fin rays (*Figure 1A*). We use the following terminology to describe the subsequent experiments: segment 0 is the fin ray segment through which we amputate (at 50% of its length), and segments −1, −2, and −3 are located further proximally (*Figure 1A*). Note that only segment 0 is mechanically affected by the amputation injury. We have

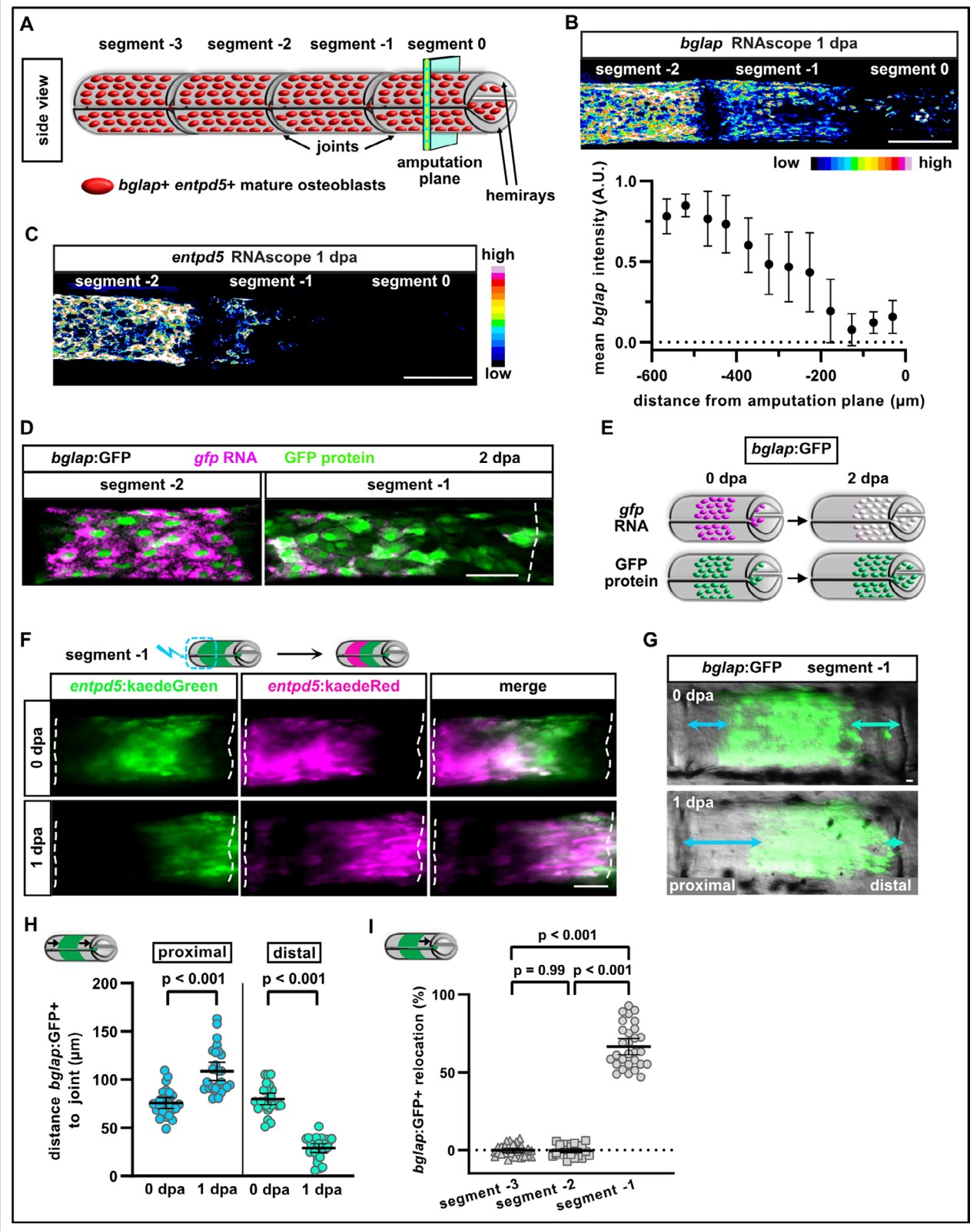

**Figure 1.** Osteoblasts dedifferentiate and migrate in response to fin amputation. (**A**) Schematic presentation of one fin ray with mature osteoblasts expressing *bglap* and *entpd5* lining the hemirays. (**B**) RNAscope in situ analysis of *bglap* expression in segments 0, –1, and –2 at 1 day post amputation (dpa). Expression of *bglap* gradually decreases towards the amputation plane. N (experiments)=1, n (fins)=5, n (rays)=15. Error bars represent SEM. Scale bar, 100 µm. (**C**) RNAscope in situ detection of *entpd5* expression in segments 0, –1, and –2 at 1 dpa. Scale bar, 100 µm. (**D**) In *bglap*:GFP fish,

*Figure 1 continued on next page*

*Figure 1 continued*

expression of *gfp* RNA is downregulated in segment –1 at 2 dpa, while GFP protein persists. Segment –2 shows overlap of *gfp* RNA and GFP protein (note that GFP protein is enriched in the nuclei). Dashed lines indicate segment border. Distal to the right. Scale bar, 50 µm (**E**) Scheme illustrating the use of *bglap*:GFP transgenics for analysis of osteoblast migration via relocation of cells that are negative for *gfp* RNA, but positive for GFP protein. (**F**) At 0 dpa, *entpd5*:Kaede positive osteoblasts in the proximal half of segment –1 were photoconverted from *entpd5*:kaedeGreen to *entpd5*:kaedeRed. Repeated imaging reveals distal relocation of photoconverted osteoblasts by 1 dpa. Dashed lines indicate segment borders. Distal to the right. Scale bar, 50 µm. (**G**) *bglap*:GFP is expressed in a subset of osteoblasts in the centre of each segment and absent around the joints. Repeated imaging reveals relocation of GFP+ cells towards the distal joint in segment –1 by 1 dpa. Distal to the right. Scale bar, 10 µm. (**H**) Quantification of the distance between the bulk of *bglap*:GFP+ cells and joints in segment –1 at 0 and 1 dpa. At 1 dpa, the proximal distance is increased, while the distal distance is reduced. N (experiments)=3, n (fins)=26, n (rays)=26. Error bars represent 95% CI. Unpaired *t*-test. (**I**) Quantification of *bglap*:GFP+ bulk migration in segment –3, –2, and –1 at 1 dpa. 100% indicates full crossing of the distance to the respective joint. Osteoblasts migrate distally in segment –1. N (experiments)=2, n (fins)=19, n (rays)=29. Error bars represent 95% CI. Mann Whitney test.

The online version of this article includes the following source data and figure supplement(s) for figure 1:

**Source data 1.** Data of experiments shown in *Figure 1B, H and I*.

**Figure supplement 1.** Osteoblasts upregulate dedifferentiation markers in response to fin amputation.

previously shown that osteoblasts dedifferentiate in response to fin amputation, that is they revert from a mature, non-proliferative state into an undifferentiated progenitor-like state, which includes loss of *bglap* expression and upregulation of the preosteoblast marker *runx2* (*Knopf et al., 2011*; *Geurtzen et al., 2014*). Using RNAscope in situ hybridisation, we can now show that downregulation of *bglap* occurs in a graded manner and that *entpd5* expression is similarly downregulated during dedifferentiation (*Figure 1B and C*). At 1 day postamputation (1 dpa), expression of *entpd5* and *bglap* remains high in segment –2, but gradually decreases towards the amputation plane and is almost entirely absent from segment 0, with *entpd5* downregulation being more pronounced (*Figure 1B and C*). While RNA expression of these genes is downregulated within hours after injury, GFP or Kaede fluorescent proteins (FPs) expressed in *bglap* or *entpd5* reporter transgenic lines persist for up to 3 days, even though transgene transcription is shut down rapidly as well (*Knopf et al., 2011*). We can confirm these earlier findings using the more sensitive RNAscope in situs. In *bglap*:GFP transgenics at 2 dpa, *gfp* RNA and GFP protein colocalised to the same cells in segment –2, where osteoblasts do not dedifferentiate (*Figure 1D*). In contrast, in the distal segment –1 GFP protein was present, but barely any *gfp* transcript could be detected (*Figure 1D*). Thus, persistence of GFP in *bglap*:GFP transgenics can be used for short-term tracing of dedifferentiated osteoblasts in zebrafish (*Figure 1E*). At 1 dpa, *bglap*:GFP+ cells upregulated expression of the preosteoblast marker *runx2a* and of *cyp26b1*, an enzyme involved in RA signalling (*Blum and Begemann, 2015*), which regulates dedifferentiation (*Figure 1—figure supplement 1A,B*). Both markers were exclusively upregulated in segment –1 and segment 0 at 1 dpa, but were absent in segment –2. Together, these data show that osteoblasts in segment –1 and segment 0 lose expression of mature markers and gain expression of dedifferentiation markers.

We have previously provided evidence that osteoblasts close to the amputation plane do not only dedifferentiate, but also relocate towards the injury site after fin amputation. To trace osteoblasts, we used the transgenic line *entpd5*:kaede (*Geurtzen et al., 2014*), in which Kaede fluorescence can be converted from green to red by UV light (*Ando et al., 2002*). We photoconverted osteoblasts in the proximal half of segment –1, while osteoblasts in the distal half remained green (*Figure 1F*). At 1 dpa, red osteoblasts were found

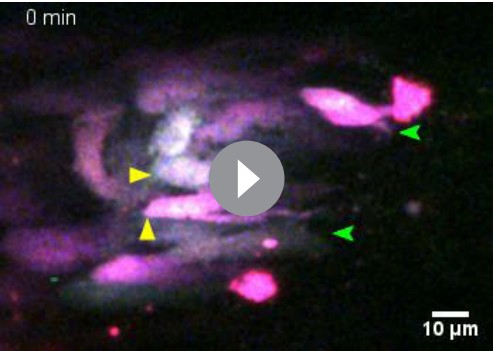

**Video 1.** Osteoblast migration. Live imaging of osteoblasts at 1 day post amputation (dpa) in segment –1 (distal to the right) in a double transgenic line expressing *bglap*:GFP and *entpd5*:kaede. Partial conversion of kaedeGreen into kaedeRed results in a different colouring for each cell. Stationary yellow arrowheads highlight the migration of cell bodies towards the amputation plane and retraction of the rear ends. Additionally, several cells can be observed forming long protrusions extending distally (green arrowheads). Distal to the right. Scale bar, 10 µm.
https://elifesciences.org/articles/77614/figures#video1

in the distal half (*Figure 1F*), showing that photoconverted osteoblasts had relocated distally. To quantify osteoblast relocation, we used the *bglap*:GFP transgenic line (*Knopf et al., 2011*). In adult non-injured fin rays, *bglap*:GFP is expressed in a subset of mature osteoblasts in the centre of every ray segment (*Figure 1G*). The restriction of *bglap*:GFP+ cells to the segment centre results in a zone devoid of *bglap*:GFP+ cells at both ends of a segment, and its invasion by GFP+ cells can be used as a read-out for bulk migration of osteoblasts after fin injury. Within 1 dpa, the distance between the GFP+ cells and the distal joint was reduced in segment –1, while at the proximal joint, the distance increased (*Figure 1G and H*). Therefore, osteoblasts did not spread out across the segment, but were relocated directionally towards the amputation plane. We also analysed if osteoblasts in more proximal segments (further away from the amputation plane) move distally and quantified their relocation. 100% relocation indicates that the distal front of the GFP+ bulk of cells has reached the respective distal joint. At 1 dpa, osteoblasts in segment –1 relocated distally, while no movement could be detected in segments –3 and –2 (*Figure 1I*). In summary, osteoblasts respond to fin amputation by dedifferentiation (detected by downregulation of transcription of differentiation markers, as well as upregulation of preosteoblast markers) and by movement towards the amputation plane, as detected by relocation of GFP+ cells (*Figure 1E*). Dedifferentiation and migration occur in a region encompassing about 350 μm proximal to the amputation plane, comprising the amputated segment 0 and the adjacent segment –1.

To observe osteoblast behaviour at single cell resolution in live fish after fin amputation, we generated double transgenic fish expressing *bglap*:GFP (*Knopf et al., 2011*) and *entpd5*:kaede in mature osteoblasts. Expression of two fluorescent proteins increased signal for repeated imaging, and partial photoconversion of kaedeGreen into kaedeRed resulted in a different colouring for each osteoblast, which facilitated tracking of morphology changes at single cell resolution. Live imaging revealed the formation of long protrusions, lasting for at least 2 hr and extending towards the amputation site, and the directed movement of cell bodies relative to their surroundings (*Video 1*, *Figure 2—figure supplement 1A*). We conclude that osteoblasts relocate by active migration. Migrating cells typically possess an identifiable cell front and a cell rear along an axis approximately aligned with the direction of locomotion. We found that *bglap*:GFP+ osteoblasts changed their shape after amputation. In a mature, non-injured segment, osteoblasts were roundish, and they retained this morphology after amputation in segment –3 and segment –2 at 1 dpa, as determined by a width/length ratio of ~0.4 (*Figure 2A and B*, *Figure 2—figure supplement 1B*). In contrast, at 1 dpa osteoblasts in segment –1 displayed an elongated shape (width/length ratio ~0.2) and they presented long extensions (*Figure 2A and B*). The elongation of osteoblasts occurred in alignment with the proximodistal axis of the fin, as evident by an angle reduction between this axis and the long axis of the osteoblasts (*Figure 2C*). Orientation of osteoblasts along the proximodistal axis could first be detected at 12 hr postamputation (hpa) (*Figure 2—figure supplement 1C*), while elongation was first observed at 15 hpa (*Figure 2—figure supplement 1D*). We interpret the elongation and orientation along the proximodistal axis and the formation of long-lived protrusions along this axis as events that prime osteoblasts for directed active migration towards the injury.

Fins grow by the addition of new segments distally, and in the distal-most, youngest segment of non-injured fins, *bglap*:GFP is not expressed (*Knopf et al., 2011*). Similarly, in the regenerating fins increasing numbers of GFP+ cells can be detected in more proximally located, older segments, reflecting the progressive differentiation of osteoblasts with time (*Figure 2—figure supplement 2A*). In less mature segments, the pre-osteoblast marker Runx2 can be detected, but its expression does not overlap with *bglap*:GFP expression in mature osteoblasts (*Figure 2—figure supplement 2B*). Importantly, in newly regenerated segments that start to upregulate *bglap* expression, all *bglap*:GFP+ cells appear in the centre of the segments; we could not observe GFP+ cells within joints (*Figure 2—figure supplement 2A*). This suggests that *bglap*:GFP+ osteoblasts from older segments do not migrate into less mature segments during formation of new segments in the course of fin growth. Rather, the mature osteoblast population in a segment arises via differentiation of osteoblasts at the position within the segment at which they were formed during segment addition. In contrast, within 2 days after fin amputation, *bglap*:GFP+ cells appeared in the fin stump within the joint between segment –1 and segment 0 (*Figure 2D*, yellow arrowhead), indicating that GFP+ osteoblasts from segment –1 crossed the joint during their migration towards the amputation plane. Thus, migration of mature osteoblasts observed after fin amputation or bone fracture (*Geurtzen et al., 2014*) appears

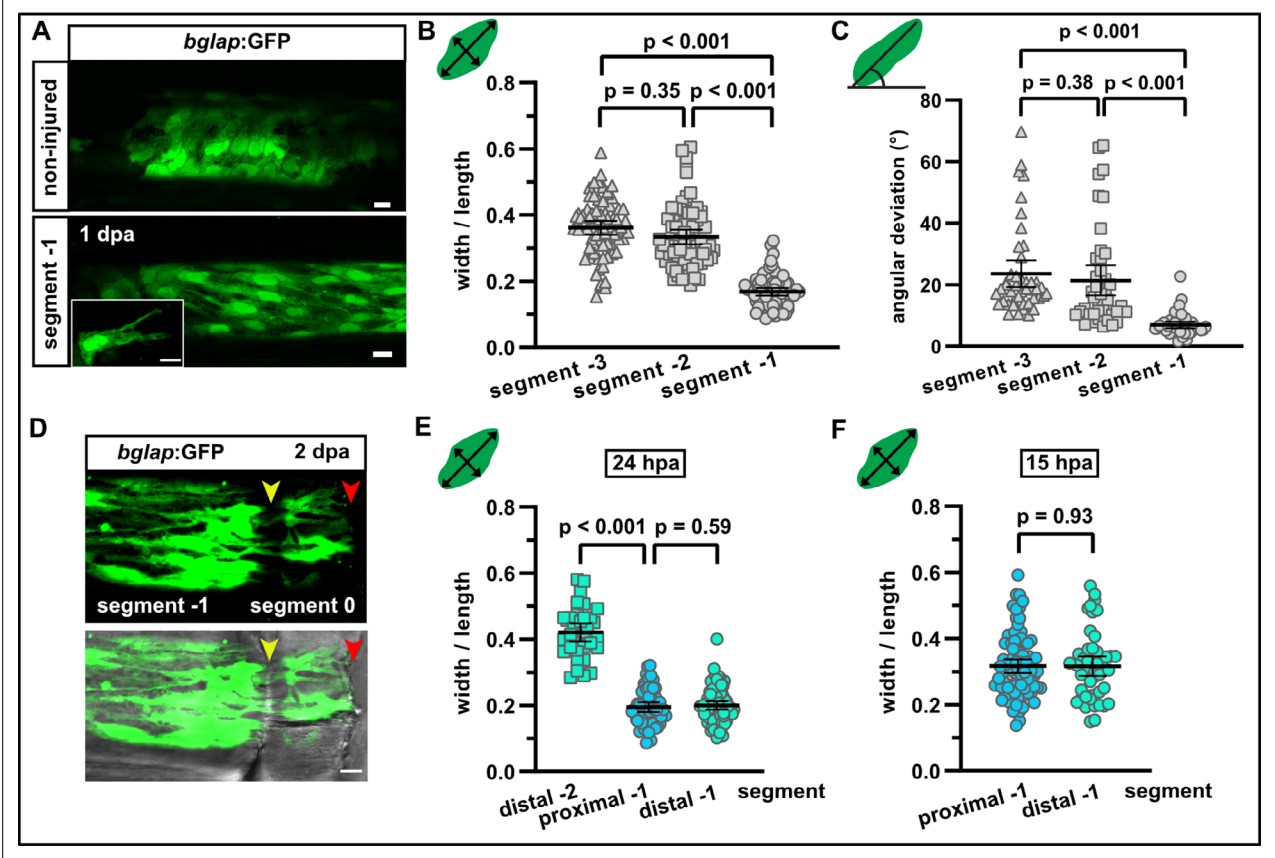

**Figure 2.** Osteoblasts elongate and orient along the proximodistal axis of the fin in response to fin amputation. (**A**) *bglap*:GFP+ osteoblast morphology in a non-injured fin (upper panel) and in segment −1 at 1 day post amputation (dpa) (lower panel). Scale bar, 10 μm. (**B**) Quantification of *bglap*:GFP+ osteoblast roundness as width/length ratio at 1 dpa. Osteoblasts are more elongated in segment −1. N (experiments)=1, n (fins)=5, n (rays)=5, n (cells)=72. Error bars represent 95% CI. Kruskal-Wallis test. (**C**) Quantification of *bglap*:GFP+ osteoblast orientation as angular deviation from the proximodistal axis at 1 dpa. In segment −1, osteoblasts display increased alignment along the axis. N (experiments)=1, n (fins)=5, n (rays)=5, n (cells)=44. Error bars represent 95% CI. Kruskal-Wallis test. (**D**) At 2 dpa, *bglap*:GFP+ osteoblasts can be observed within the distal joint of segment −1, and GFP+ protrusions can be seen spanning the joint, indicating that osteoblasts migrate through the joint. Yellow arrowheads, joints; red arrowheads, amputation plane. Scale bar, 10 μm. (**E**) Osteoblast roundness in distal and proximal parts of segments (40% of total segment length from the respective end) at 24 hpa. No difference within segment −1 can be detected. N (experiments)=1, n (fins)=5, n (rays)=5, n (cells)=34 (segment −2), 55 (proximal segment −1), 78 (distal segment −1). Error bars represent 95% CI. Mann-Whitney test. (**F**) Osteoblast roundness in distal and proximal parts of segment −1 at 15 hpa. No difference within segment −1 can be detected. N (experiments)=1, n (fins)=6, n (rays)=15, n (cells)=82 (proximal), 49 (distal). Error bars represent 95% CI. Mann-Whitney test. The observed relative difference is 0.3%, the calculated smallest significant difference 13%, which is smaller than what we observe between segment −2 and segment −1 (**B**, 54%).

The online version of this article includes the following source data and figure supplement(s) for figure 2:

**Source data 1.** Data of experiments shown in *Figure 2B, C, E and F*, and *Figure 2—figure supplement 1B, C and D*.

**Figure supplement 1.** Osteoblasts elongate and orient along the proximodistal axis of the fin in response to fin amputation.

**Figure supplement 2.** The *bglap*:GFP transgenic reporter is specific for a subset of differentiated osteoblasts.

to represent a specific early response to injury, while addition of new bony segments at the distal tip of growing fins during ontogeny or regeneration does not involve migration of differentiated osteoblasts.

Since we found that differentiation markers are gradually downregulated towards the amputation plane at 1 dpa (*Figure 1B and C*), we wondered whether a similar gradient can be observed for cell morphology changes. However, osteoblasts were elongated to the same extent at the proximal and distal end of segment −1 at 24 hpa (*Figure 2E*). As elongation was first observed at 15 hpa (*Figure 2—figure supplement 1D*), we also analysed osteoblasts in proximal and distal regions of segment −1 at this time point. Yet, no morphological differences in osteoblasts of proximal and distal regions of segment −1 were detected at 15 hpa (*Figure 2G*). We conclude that dedifferentiation and cell shape

changes are responses of osteoblasts to injury that occur in the same cells, yet the magnitude of these responses can vary. While cell shape change is a binary response that affects all osteoblasts in the responsive zone equally, dedifferentiation is a graded process.

## Actomyosin, but not microtubule dynamics is required for osteoblast cell shape changes and migration

One of the primary forces facilitating cell shape changes and cell motility is the myosin-associated actin cytoskeleton (*Chugh and Paluch, 2018*). To interfere with the treadmilling of actin microfilaments (F-actin), we used cytochalasin D, which binds to F-actin and disturbs its turnover (*Brown and Spudich, 1979*; *Brown and Spudich, 1981*). Drug treatment strongly impaired osteoblast elongation in segment –1 at 1 dpa, while it did not affect osteoblast morphology in segment –2 and segment –3 (*Figure 3A*). Concomitantly, treatment with cytochalasin D also resulted in reduced alignment of osteoblasts along the proximodistal axis in segment –1, but did not affect the orientation of osteoblasts in segment –2 and segment –3 (*Figure 3B*). Interference with actin dynamics also significantly reduced the bulk migration of *bglap*:GFP+ cells in segment –1 (*Figure 3C*). To interfere with another hallmark of a dynamic cytoskeleton, the contractility of the actomyosin network, we injected fish with blebbistatin, an inhibitor of myosin II ATPase (*Kovács et al., 2004*). The drug did not affect osteoblast cell shape in segments –3 and –2, but impaired the elongation and reorientation of osteoblasts along the proximodistal axis in segment –1 (*Figure 3D and E*). Concomitantly, bulk osteoblast migration was reduced (*Figure 3C*).

Due to osteoblast dedifferentiation, at 1 dpa *bglap* RNA expression in segment –1 is reduced to ~50% compared to the expression in segment –2 (*Figure 3—figure supplement 1A*). Interestingly, downregulation of *bglap* expression was not affected by either cytochalasin D or blebbistatin treatment (*Figure 3F*), indicating that their effect on cell shape change and migration is not secondary to impaired dedifferentiation. Likewise, upregulation of Runx2 at 2 dpa was not diminished upon cytochalasin D treatment (*Figure 3—figure supplement 1B*). Yet, regenerative growth was reduced at 3 dpa (*Figure 3—figure supplement 1C*).

Besides the actomyosin network, microtubules (MT) are an important force to drive cell shape changes and cell motility (*Etienne-Manneville, 2013*). To analyse a potential role of MT in the amputation-induced migration of osteoblasts, we treated fish with nocodazole, a drug that disrupts MT assembly/disassembly (*Florian and Mitchison, 2016*), using a regime which was shown to be effective in zebrafish (*Poss et al., 2004*). At 1 dpa, we could not detect a difference in the extent of migration in drug-treated fish compared to controls (*Figure 3G*). Together, these data indicate that actomyosin, but not microtubule dynamics is required for injury-induced osteoblast migration.

## Osteoblast migration is independent of osteoblast dedifferentiation

We next wondered whether dedifferentiation is a prerequisite for osteoblasts to change their shape and to migrate. We have previously shown that NF-κB signalling negatively regulates osteoblast dedifferentiation (*Mishra et al., 2020*). We induced expression of an inhibitor of NF-κB signalling (IkB super repressor [IkBSR]); (*Van Antwerp et al., 1996*) specifically in osteoblasts, using the tamoxifen-inducible Cre line *OlSp7*:CreERT2-p2a-mCherry[tud8] (*osx*:CreER) (*Knopf et al., 2011*) and an ubiquitously expressed responder line *hsp70l*:loxP Luc2-myc Stop loxP nYPet-p2a-IκBSR, *cryaa*:AmCyan[ulm15Tg], which expresses IkBSR and nYPet after recombination (*hs*:Luc to nYPet IκBSR). NF-kB signalling activation was induced by expressing constitutively active human IKK2 (IkB kinase) and nuclear localised BFP after recombination in osteoblasts using *hsp70l*:loxP Luc-myc Stop loxP IKKca-t2a-nls-mTagBFP2-V5, cryaa:AmCyan[ulm12Tg] (*hs*:Luc to IKKca BFP) fish (*Mishra et al., 2020*). Expression of GFP driven by the Cre-responder line *hsp70l*:loxP DsRed2 loxP nlsEGFP[tud9] (*hs*:R to G) (*Knopf et al., 2011*) served as a negative control. Using these tools, we have previously shown that activation of NF-κB-signalling in osteoblasts inhibits their dedifferentiation, while its suppression promotes osteoblast dedifferentiation (*Mishra et al., 2020*). We used the same setup to ask whether NF-κB signalling also regulates osteoblast migration. Mosaic recombination allowed us to compare recombined and non-recombined cells within the same segment. Analysis of cell shape in the control line *hs*:R to G revealed elongation of both non-recombined and recombined cells in segment –1 compared to segment –2 (*Figure 4A*). Neither pathway inhibition by expression of IkBSR nor forced activation by expression

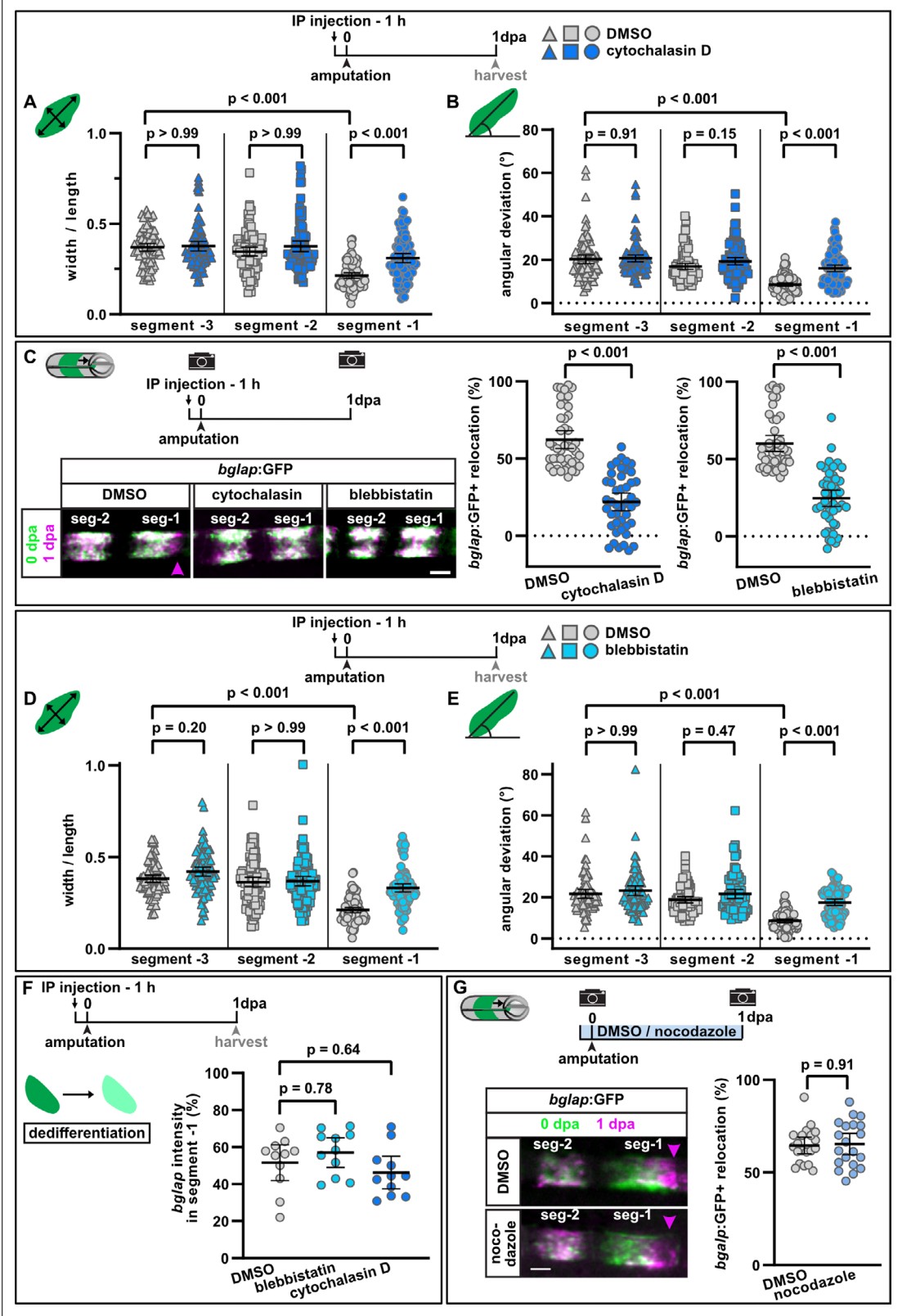

**Figure 3.** Interference with actomyosin, but not microtubule dynamics inhibits osteoblast cell shape changes and migration. (**A**) Osteoblast roundness at 1 day post amputation (dpa). Inhibition of actin dynamics with cytochalasin D does not alter osteoblast cell shape in segments –3 and –2, but cell elongation in segment –1 is inhibited. N (experiments)=3, n (fins)=15, n (rays)=15, n (cells)=87. Error bars represent 95% CI. Kruskal-Wallis test. (**B**) Osteoblast orientation at 1 dpa. Cytochalasin D does not alter osteoblast orientation in segments –3 and –2, but alignment along the proximodistal

*Figure 3 continued on next page*

*Figure 3 continued*

axis in segment –1 is impaired. N (experiments)=3, n (fins)=15, n (rays)=15, n (cells)=93. Error bars represent 95% CI. Kruskal-Wallis test. (**C**) Both cytochalasin D and blebbistatin treatment impair bulk osteoblast migration. Images show overlay of 0 dpa (green) and 1 dpa (pink) pictures, with the pink arrowhead indicating relocation of osteoblasts in controls, where no signal overlap is observed at the distal side. Graph: 100% indicates full crossing of the distance to the respective joint at 1 dpa. Cytochalasin D: N(experiments)=3, n (fins)=22, n (rays)=44; blebbistatin: n (fins)=24, n (rays)=48, appertaining controls have the same n. Error bars represent 95% CI. Mann-Whitney test. Scale bar, 100 μm. (**D**) Osteoblast roundness at 1 dpa. Inhibition of myosin activity with blebbistatin does not alter osteoblast cell shape in segments –3 and –2, but cell elongation in segment –1 is inhibited. N (experiments)=3, n (fins)=15, n (rays)=15, n (cells)=89. Error bars represent 95% CI. Kruskal-Wallis test. (E) Osteoblast orientation at 1 dpa. Blebbistatin does not alter osteoblast orientation in segments –3 and –2, but alignment along the proximodistal axis in segment –1 is impaired. N (experiments)=3, n (fins)=15, n (rays)=15, n (cells)=77. Error bars represent 95% CI. Kruskal-Wallis test. (F) Neither cytochalasin nor blebbistatin treatment affects osteoblast dedifferentiation. Plotted are the 1 dpa *bglap* RNAscope signal intensity levels in segment –1 relative to those in the same ray in segment –2. N (experiments)=1, n (fins)=6, n (rays)=12. Error bars represent 95% CI. Kruskal-Wallis test. The observed relative difference is 3% (blebbistatin) and 2% (cytochalasin), the calculated smallest significant difference for both assays is 9%, which is smaller than what we observe after retinoic acid treatment (*Figure 4B*, 63%). (G) Inhibition of microtubule dynamics with nocodazole does not affect osteoblast migration. Images show overlay of 0 dpa (green) and 1 dpa (pink) pictures. N (experiments)=1, n (fins)=10, n (rays)=20. Error bars represent 95% CI. Unpaired *t*-test. The observed relative difference is 1%, the calculated smallest significant difference is 24% and thus smaller than what we observe in actomyosin treatment regimens (C, 60–63%). Scale bar, 100 μm.

The online version of this article includes the following source data and figure supplement(s) for figure 3:

**Source data 1.** Data and effect size calculations of experiments shown in *Figure 3A, B, C, D, E, F and G*, and *Figure 3—figure supplement 1A B and C*.

**Figure supplement 1.** Interference with actomyosin dynamics does not affect upregulation of the preosteoblast marker Runx2 in dedifferentiating osteoblasts, but reduces growth of the regenerate.

of IKKca affected osteoblast elongation in segment –1. We conclude that NF-κB signalling regulates osteoblast dedifferentiation, but not the cell shape changes associated with osteoblast migration.

RA signalling inhibits osteoblast dedifferentiation downstream of NF-κB signalling (*Blum and Begemann, 2015*; *Mishra et al., 2020*). Treatment with RA impaired osteoblast dedifferentiation after amputation, as *bglap* RNA remained at higher levels in segment –1 (*Figure 4B*). In contrast, osteoblast elongation and alignment along the proximodistal axis in segment –1 at 1 dpa were not affected (*Figure 4C and D*). Similarly, RA did not affect bulk osteoblast migration towards the amputation plane (*Figure 4E*).

Together, the data on manipulation of actomyosin dynamics, NF-κB, and RA signalling suggest that osteoblast dedifferentiation and migration are independently regulated, as one osteoblast injury response can be impaired without affecting the other. Furthermore, they indicate that osteoblast dedifferentiation is not a prerequisite for migration.

## The complement system is required for osteoblast migration in vivo

As shown above, we found that osteoblast cell shape changes and migration are directed towards the amputation site. Directional cell migration is usually initiated in response to extracellular cues such as chemokines or signals from the extracellular matrix (*Swaney et al., 2010*). As part of the innate immune system, the complement system is activated immediately after injury, and activated complement factors can act as chemoattractants for osteoblasts in vitro (*Ignatius et al., 2011a*). We thus asked whether the complement system regulates osteoblast migration in vivo. The zebrafish genome contains homologues of all fundamental mammalian complement components (*Boshra et al., 2006*; *Zhang and Cui, 2014*), including the central components *c3* and *c5* and the corresponding receptors *c3aR* and *c5aR1*, respectively. A second *c5a* receptor, *c5aR2*, has so far only been characterised in mammals, and sequence database interrogation indicates that it is absent in zebrafish. RNAscope in situ analysis revealed that *c5aR1* is expressed in mature osteoblasts (*Figure 5A*), as well as in other cells of the fin (*Figure 5—figure supplement 1A*). Similarly, quantitative RT-PCR (qRT-PCR) on GFP+ and GFP– cells sorted from 1 dpa *bglap*:GFP fins revealed that *c5aR1* is expressed in both fractions, and thus in osteoblasts and other cell types (*Figure 5—figure supplement 1B*). Likewise, expression of *c3aR1* could be detected in both fractions (*Figure 5—figure supplement 1B*). As measured by qRT-PCR, transcription of the complement factor precursor *c5* and the six *c3a* precursor paralogues could readily be detected in the liver, but was minimal in non-injured fins ($10^3$–$10^6$-fold less than in the liver, *Figure 5B*). Yet, we could observe an upregulation of *c5* and *c3a.5* expression in the fin at 6 hpa (*Figure 5B*). Thus, while the majority of complement components that are available for injury-induced

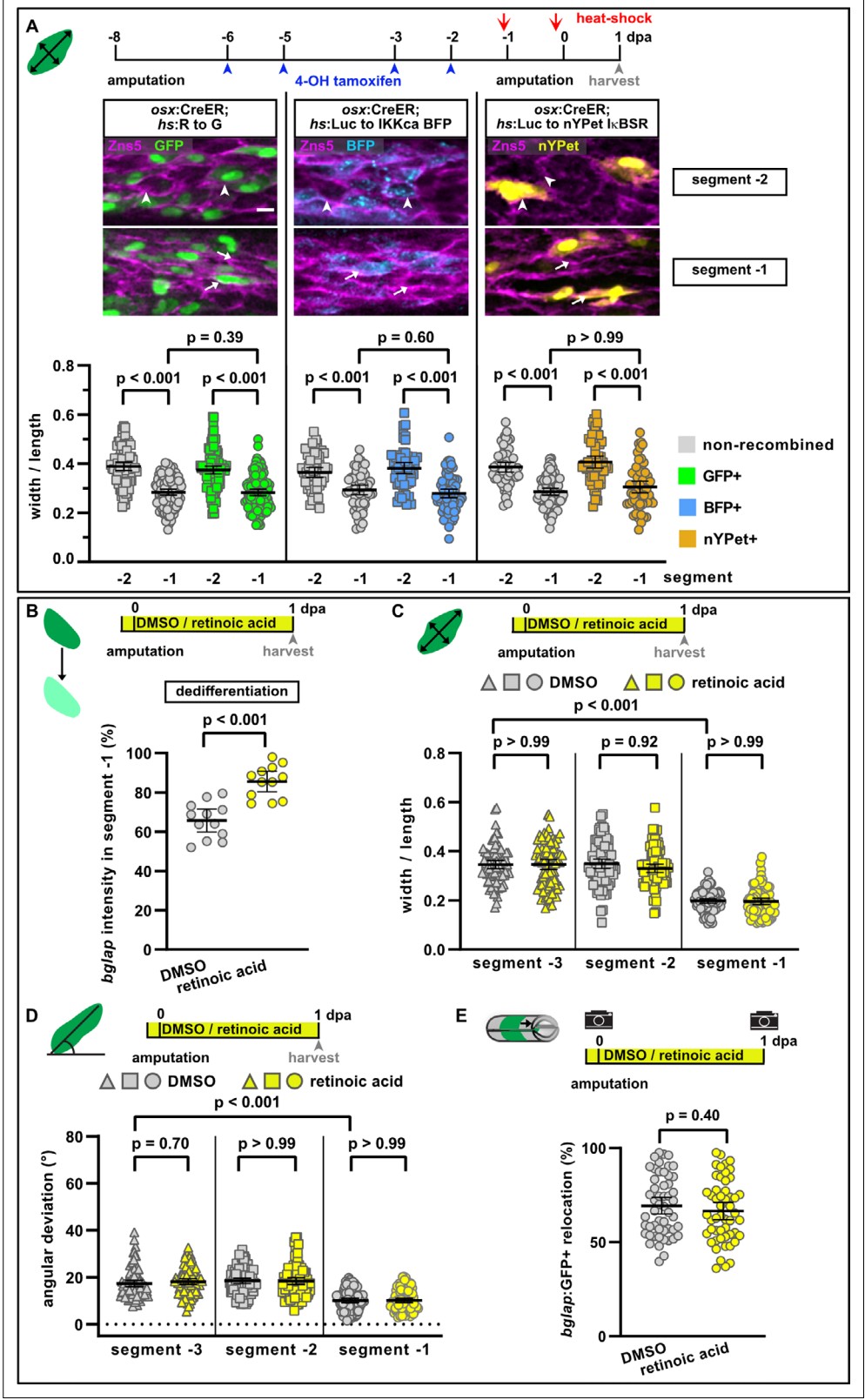

**Figure 4.** NF-$\kappa$B and retinoic acid signalling regulate osteoblast dedifferentiation but not migration. (**A**) Mosaic recombination in *osx*:CreER; *hs*:R to G, *osx*:CreER; *hs*:Luc to IKKca BFP and *osx*:CreER; *hs*:Luc to nYPet I$\kappa$BSR fish. Zns5 labels the membrane of osteoblasts. White arrowheads highlight roundish osteoblasts, white arrows elongated osteoblasts. Osteoblast roundness in recombined and non-recombined osteoblasts in segment –2

*Figure 4 continued on next page*

*Figure 4 continued*

and segment –1 at 1 day post amputation (dpa) is plotted. Recombined osteoblasts expressing IKKca (marked by BFP) or I κ BSR (marked by nYPet) elongate in segment –1 to a similar extent as osteoblasts expressing the negative control GFP. N (experiments)=1, R to G: n (fins)=3, n (rays)=10, n (cells)=406; IKKca: n (fins)=4, n (rays)=7, n (cells)=244; I κ BSR: n (fins)=8, n (rays)=18, n (cells)=246. Error bars represent 95% CI. Kruskal-Wallis test. Scale bar, 10 μm. (**B**) Treatment with retinoic acid (RA) inhibits osteoblast dedifferentiation measured as *bglap* RNAscope intensity at 1 dpa in segment –1 relative to the intensity in segment –2 in the same rays. N (experiments)=1, n (fins)=6, n (rays)=12. Error bars represent 95% CI. Unpaired *t*-test. (**C**) Osteoblast roundness at 1 dpa. RA treatment does neither alter osteoblast cell shape in segments –3 and –2, nor elongation in segment –1. N (experiments)=2, n (fins)=10, n (rays)=10, n (cells)=83. Error bars represent 95% CI. Kruskal-Wallis test. The observed relative difference in segment –1 is 2%, the calculated smallest significant difference is 7%, which is smaller than what we observe in cytochalasin treatment regimens (***Figure 3A***, 22%). (**D**) Osteoblast orientation at 1 dpa. RA treatment does not affect osteoblast orientation in segments –3 and –2, nor alignment along the proximodistal axis in segment –1. N (experiments)=2, n (fins)=10, n (rays)=10, n (cells)=89. Error bars represent 95% CI. Kruskal-Wallis test. The observed relative difference in segment –1 is 1%, the calculated smallest significant difference is 7%, which is smaller than what we observe in cytochalasin treatment regimens (***Figure 3B***, 32%). (**E**) RA treatment does not affect bulk migration of osteoblasts towards the amputation plane in segment –1. N (experiments)=3, n (fins)=26, n (rays)=52. Error bars represent 95% CI. Unpaired *t*-test. The observed relative difference is 2%, the calculated smallest significant difference is 22%, which is smaller than what we observe in actomyosin treatment regimens (***Figure 3C*** 60–63%).

The online version of this article includes the following source data for figure 4:

**Source data 1.** Data and effect size calculations of experiments shown in ***Figure 4A, B, C, D and E***.

---

activation in the fin are probably produced in the liver and delivered to the fin by the circulation, injury-induced local expression in the fin might contribute to activation of the system as well.

Treatment with W54011, a specific C5aR1 antagonist (***Sumichika et al., 2002***), impaired osteoblast elongation in segment –1 at 1 dpa without affecting osteoblast cell shape in segments –2 and –3 (***Figure 5C***). As also the complement factor C3a can act as chemoattractant after injury (***Huber-Lang et al., 2013***), we treated fish with SB290157, a specific antagonist of the complement receptor C3aR (***Ames et al., 2001***). Osteoblast elongation was impaired in segment –1 at 1 dpa, while cell shape in segment –2 and segment –3 was not affected (***Figure 5D***). Both drugs also significantly reduced bulk osteoblast migration (***Figure 5E***). To confirm these findings, we repeated the migration assay with PMX205, another C5aR1 antagonist (***March, 2004***; ***Jain et al., 2013***), which also reduced osteoblast migration at 1 dpa (***Figure 5E***). Together, these data strongly suggest that the complement system regulates injury-induced directed osteoblast migration in vivo.

Interestingly, none of the complement inhibitors affected osteoblast dedifferentiation as quantified by reduction of *bglap* RNA expression in segment –1 (***Figure 5F***). This supports the view that osteoblast dedifferentiation and migration are independent responses of osteoblasts to injury. Complement components can also contribute to the regulation of cell proliferation and tissue regeneration (***Mastellos and Lambris, 2002***). In the non-injured fin, *bglap*:GFP+ osteoblasts are non-proliferative, but upon amputation osteoblasts proliferate at 2 dpa (***Figure 5—figure supplement 2A ,B***). Proliferation is restricted to segment –1 and segment 0 (***Figure 5—figure supplement 2C***), and RNAscope in situ analysis of *bglap* expression revealed that the majority of EdU+ osteoblasts have strongly downregulated *bglap* (***Figure 5—figure supplement 2D***). Inhibition of C5aR1 with PMX205 had no effect on osteoblast proliferation in segment –1 at 2 dpa (***Figure 5—figure supplement 3A***). Furthermore, upregulation of Runx2 was not changed by PMX205 treatment (***Figure 5—figure supplement 3B***), and regenerative growth was not affected in fish treated with either W54011, PMX205, or SB290157 (***Figure 5—figure supplement 3C***). We conclude that the complement system specifically regulates injury-induced osteoblast migration, but not osteoblast dedifferentiation or proliferation in zebrafish.

Activation of the complement system is a general early response after injury that also occurs in wounds that heal, but do not trigger structural regeneration of the kind we observe after fin amputation. Thus, we wondered whether osteoblast migration can be triggered by injuries that do not induce structural regeneration. Lesions of the interray skin quickly heal (***Chablais and Jazwinska, 2010***; ***Chen et al., 2015***) but do not trigger bone regeneration. Yet, such wounds can induce signals that are sufficient to trigger structural regeneration in a missing tissue context (***Owlarn et al., 2017***). We found that skin injuries induced close to the ray bone did not induce osteoblast migration, neither off the

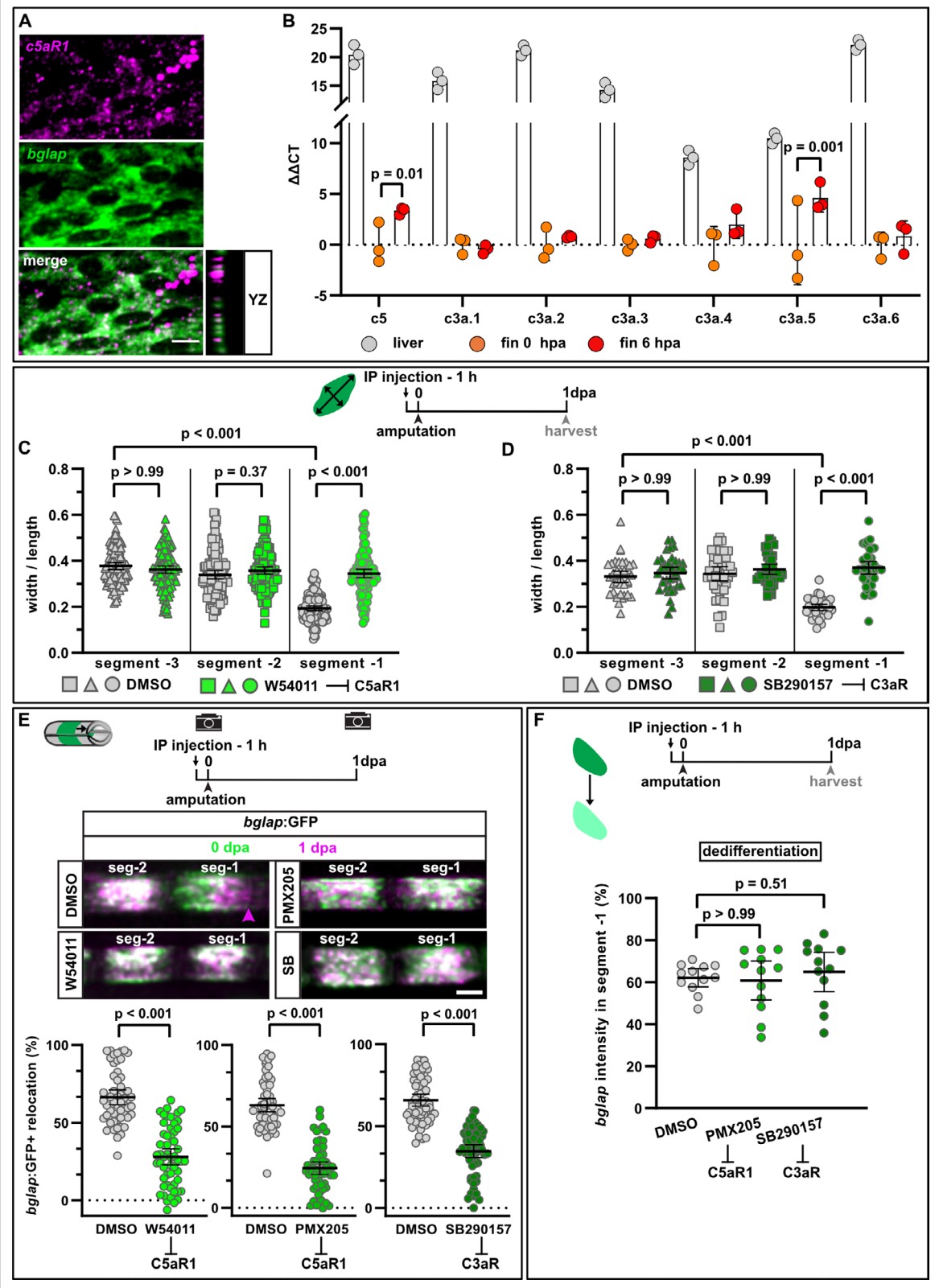

**Figure 5.** Complement system signalling regulates osteoblast cell shape changes and migration in vivo. (**A**) RNAscope in situ detection of *c5aR1* expression in *bglap* expressing osteoblasts in a non-injured fin segment. YZ section shows localisation of both RNAs in the same cell layer. Scale bar, 10 µm. (**B**) Expression of complement factors detected by qRT-PCR in the liver, fins at 0 hpa and 6 hpa. Plotted are the ΔΔCT values relative to the 0 hpa fins. Each data point represents one biological replicate. Error bars represent SD. Two-way ANOVA. (**C**) Osteoblast roundness at 1 day post amputation

*Figure 5 continued on next page*

*Figure 5 continued*

(dpa). The C5aR1 inhibitor W54011 does not alter osteoblast cell shape in segments –3 and –2, but cell elongation in segment –1 is inhibited. N (experiments)=3, n (fins)=5, n (rays)=5, n (cells)=116. Error bars represent 95% CI. Kruskal-Wallis test. (**D**) Osteoblast roundness at 1 dpa. The C3R inhibitor SB290157 does not alter osteoblast cell shape in segments –3 and –2, but cell elongation in segment –1 is inhibited. N (experiments)=1, n (fins)=5, n (rays)=5, n (cells)=38. Error bars represent 95% CI. Kruskal-Wallis test. (**E**) Inhibition of C5aR1 with either W54011 or PMX205, and inhibition of C3aR with SB290157 impairs bulk osteoblast migration. Images show overlay of 0 dpa (green) and 1 dpa (pink) pictures. N (experiments)=3, W54011: n (fins)=27, n (rays)=54; PMX-205, SB290157: n (fins)=29, n (rays)=58. Error bars represent 95% CI. Mann-Whitney tests. Scale bar, 100 μm. (**F**) Neither inhibition of C5aR1 (W54011, PMX205) nor of C3aR (SB290157) affects osteoblast dedifferentiation measured as *bglap* RNAscope intensity at 1 dpa in segment –1 relative to the intensity in segment –2 of the same rays. N (experiments)=1; n (fins)=5 (DMSO), 4 (PMX205), 6 (SB290157); n (rays)=12. Error bars represent 95% CI. Kruskal-Wallis test. The observed relative difference is 1% for PMX205 and SB290157, the calculated smallest significant differences are 6% (PMX205) and 5% (SB290157), which are smaller than what we observe after retinoic acid treatment (*Figure 4B*; 7%).

The online version of this article includes the following source data and figure supplement(s) for figure 5:

**Source data 1.** Data and effect size calculations of experiments shown in *Figure 5B, C, D, E and F*, and *Figure 5—figure supplement Figure 5—figure supplements 1B and 2B, C, D Figure 5—figure supplement 3A, B, C*.

**Figure supplement 1.** Localisation of complement receptors in the zebrafish fin.

**Figure supplement 2.** Proliferation of *bglap*:GFP+ osteoblasts.

**Figure supplement 3.** Complement system signalling is not required for osteoblast proliferation, dedifferentiation, and regenerative growth.

**Figure supplement 4.** Injuries that do not trigger structural regeneration do not attract osteoblasts.

bone into the intraray nor on the bone towards the joints (*Figure 5—figure supplement 4A*). Osteoblasts also migrate towards fractures induced in the fin ray bone (*Geurtzen et al., 2014*). Interray injury combined with bone fracture resulted in osteoblast migration towards the fracture, but osteoblasts did not migrate away from the bone matrix into the interray skin (*Figure 5—figure supplement 4B*). We conclude that generic wounding-induced signals are not sufficient to attract osteoblasts in the absence of additional prerequisites, which might include the presence of a permissive substrate for migration.

## An internal bone defect model separates generic from regeneration-specific osteoblast injury responses

In response to fin amputation, all osteoblast injury responses occur directed towards the amputation plane, that is, dedifferentiation is more pronounced distally, osteoblasts migrate distalwards, and the proliferative preosteoblast population forms distally of the amputation plane. We wondered how osteoblasts respond to injuries that occur proximal to their location. To test this, we established a fin ray injury model featuring internal bone defects. We removed hemiray segments at two locations within one ray, leaving a centre segment with two injury sites, one facing proximally, the other distally (*Figure 6A*). It was recently shown that in a similar cavity injury model a blastema forms only at the distal-facing injury, while the proximally facing site displays no regenerative growth (*Cao et al., 2021*). In our hemiray removal model, both injury sites are equally severed from the stump, and thus cut off from innervation and blood supply. Therefore, any potential differences in the injury responses at the proximal vs distal injury site cannot be explained by differences in innervation or blood circulation. Within 2 days post injury (dpi), blood flow through the centre segment was restored (*Figure 6—figure supplement 1A*, *Video 2*). At 3 dpi, Runx2+ preosteoblasts and Osterix+ committed osteoblasts accumulated almost exclusively in the defect beyond the distal injury, but not at the proximal injury (*Figure 6B - D*). Subsequently, new bone matrix formed exclusively at the distal site (*Figure 6E*). New bone matrix also formed at the distal side of the bone located proximal to the proximal bone defect (arrowhead in *Figure 6A* and data not shown), but not at the proximal injury site of the centre segment. Thus, our hemiray removal injury model reveals that accumulation of a preosteoblast population, and regeneration of bone occurs only at distal-facing wounds.

In response to regular fin amputations, the initiation of osteoblast dedifferentiation, migration, and proliferation occur prior to the accumulation of a preosteoblast population in the blastema. Thus, we asked whether any of these osteoblast injury responses occur at proximal-facing injuries in the hemiray injury model, which fail to form a blastema containing preosteoblasts. At 2 and 3 dpi, proliferation of osteoblasts was observed throughout the centre segment (*Figure 6F*). While at 2 dpi, the median of the distribution of proliferating osteoblasts along the proximodistal axis was slightly shifted

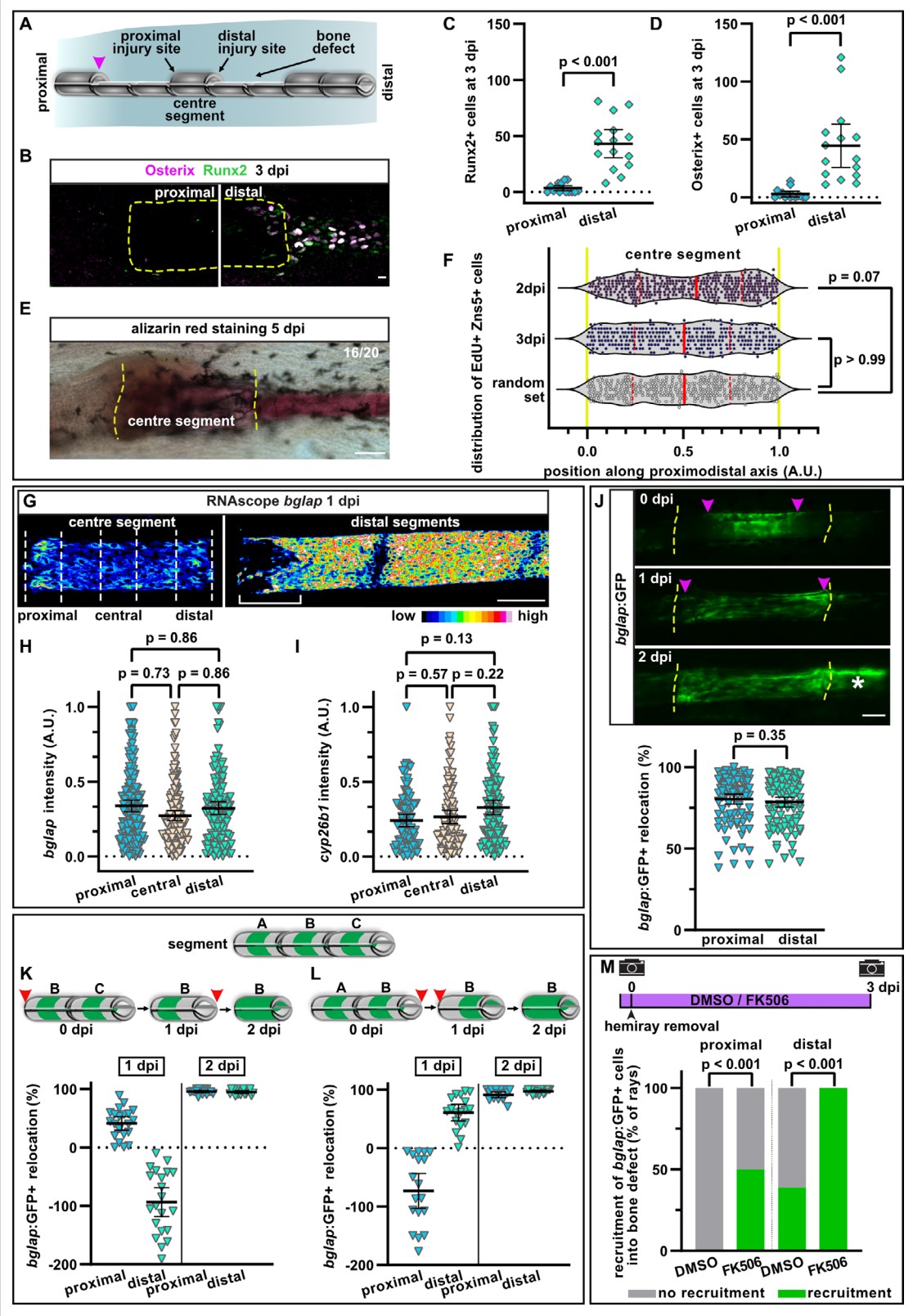

**Figure 6.** A hemiray removal injury model distinguishes generic vs regeneration-specific osteoblast injury responses. (**A**) Hemiray removal scheme creating a proximal and distal facing injury on both sides of a central intact segment. (**B**) At 3 days post injury (dpi), Runx2+ preosteoblasts and committed osteoblasts expressing Osterix have only accumulated in the bone defect beyond the distal injury site of the centre segment, but not beyond the proximal injury site, as determined by immunofluorescence. Dashed line outlines centre segment. Scale bar, 10 μm. (**C, D**) Quantification

*Figure 6 continued*

of the number of Runx2+ cells (**C**) or Osterix+ cells (**D**) located in the bone defect beyond the proximal and distal injury sites of the centre segment at 3 dpi. N (experiments)=2, n (rays)=15. Error bars represent 95% CI. Wilcoxon matched-pairs test. (**E**) At 5 dpi, mineralised bone as detected by alizarin red staining has formed beyond the distal, but not the proximal injury site of the centre segment. Dashed line indicates centre segment. n=16/20 rays with distal bone formation. Scale bar, 100 μm. (**F**) Distribution of all positions along the proximodistal axis of centre segments, where proliferating EdU+ Zns5+ osteoblasts were observed at 2 and 3 dpi. Yellow dashed lines indicate segment border. Solid red lines, median; dashed lines, quartiles. N (experiments)=1, 2 dpi: n (rays)=13, n (cells)=306; 3 dpi: n (rays)=17, n (cells)=305; random set: n (groups)=12, n (points)=360. Kolmogorov-Smirnov test. (**G**) RNAscope in situ detection of *bglap* expression in the centre segment and the two adjacent segments distal to the bone defect at 1 dpi. Dashed lines indicate the proximal, central, and distal regions of the centre segment used for quantification. Bracket indicates dedifferentiation zone in the distal segments. Scale bar, 100 μm. (**H, I**) Single cell analysis of RNAscope intensity of *bglap* (**H**) or *cyp26b1* (**I**) relative to the brightest signal in proximal, central, and distal regions of the centre segment at 1 dpi. N (experiments)=2, n (segments)=8 (*bglap*), 7 (*cyp26b1*). Error bars represent 95% CI. Holm-Sidak's multiple comparison test. (**J**) Migration of *bglap*:GFP+ osteoblasts towards both injury sites of the centre segment. Yellow dashed lines, segment borders. Pink arrowheads indicate relocation of GFP+ osteoblasts. Asterisk indicates GFP+ osteoblasts that have entered the distal bone defect. Distal to the right. N (experiments)=4, n (segments)=95. Error bars represent 95% CI. Wilcoxon matched-pairs test. Scale bar, 100 μm. (**K, L**) Sequential hemiray injuries reveal no preference for osteoblast migration in distal vs proximal directions. Relocation of *bglap*:GFP+ osteoblasts in the centre segment (segment B) is plotted. Error bars represent 95% CI. Red arrowheads indicate time points at which the adjacent hemirays (A=proximal, C=distal) were removed. Negative relocation indicates increased distance between GFP+ cells and the joint. (**K**) Removal of the proximal adjacent segment A at 0 dpi, followed by removal of the distal adjacent segment C at 1 dpi. (**L**) Removal of the distal adjacent segment C at 0 dpi, followed by removal of the proximal adjacent segment A at 1 dpi. (**M**) Migration of osteoblasts into the bone defect at 3 dpi. Treatment with FK506 induces recruitment of GFP+ cells beyond the proximal injury site and increases the number of centre segments that show recruitment of GFP+ cells beyond the distal injury site. N (experiments)=1, n (segments)=56 (DMSO), 17 (FK506). Fisher's test.

The online version of this article includes the following source data and figure supplement(s) for figure 6:

**Source data 1.** Data of experiments shown in *Figure 6C, D, F, H, I, J, K, L and M*, and *Figure 6—figure supplement 1A,C,D,E,F*.

**Figure supplement 1.** Properties of the hemiray injury model.

distally compared to a random distribution set, at 3 dpi proliferation was equally distributed along the segment (*Figure 6F*). Thus, the polarised regenerative response is not reflected by equally polarised osteoblast proliferation within the centre segment.

We next wondered whether distally-oriented bone regeneration is due to varying degrees of osteoblast dedifferentiation along the segment. We analysed the expression of *bglap* with single cell resolution in a proximal, central, and distal region of the centre segment at 1 dpi. Within such a distance (~300 μm), a gradual change in *bglap* intensity can be observed after fin amputation (*Figure 1B*) and in the segments distally of the distal bone defect in the hemiray injury model (bracket in *Figure 6G*, right panel). Yet, we could not detect differences in the extent of *bglap* expression between the regions of the centre segment (*Figure 6G*, left panel, *Figure 6H*), indicating that the proximal and distal injury sites induced osteoblast dedifferentiation. Similarly, *cyp26b1* was upregulated along the entire segment, with no enrichment at the distal injury (*Figure 6I*, *Figure 6—figure supplement 1B*). Together, these data show that osteoblast dedifferentiation occurs to a similar extent in the bone facing the proximal and distal injury.

Intriguingly, osteoblasts in the centre segment also migrated towards both amputation planes, ultimately spreading across the entire segment (arrowheads in *Figure 6J*). The magnitude of migration was similar towards both wound sites (*Figure 6J*). This suggests that at this early time point, there is no hierarchy between the injury sites, and that osteoblasts react to an initial wound signal, which is independent of the proximodistal position of the injury. Furthermore, as osteoblasts also migrate towards the proximal site where no blastema will form, it appears that osteoblasts migrate independently of whether this will be followed by bone regeneration. To further test this, we temporally separated the creation of the two bone defects flanking the centre segment (*Figure 6K and L*). We first

**Video 2.** Revascularisation of the centre segment. Live imaging of blood flow at 2 days post injury (dpi). Yellow arrowheads indicate segment borders of the centre segment. Lower ray is not injured. Distal to the right. Scale bar, 100 μm.
https://elifesciences.org/articles/77614/figures#video2

extracted only one hemiray (segment A) and analysed osteoblasts of the adjacent distal segment (segment B). Within 1 day, the distance between osteoblasts and the distal segment border increased, while proximally it decreased, revealing a directed migration of the osteoblasts towards the proximally facing injury (*Figure 6K*). We then removed the hemiray of segment C (the one located distally to the centre segment B) and analysed osteoblast locations in segment B 1 day later. The distance between osteoblasts and the proximal segment border further decreased, indicating that the osteoblasts continued to migrate towards the proximal injury (*Figure 6K*). However, at the distal side, the previously increased distance was decreased, suggesting that osteoblasts reversed their direction and migrated towards the distal injury as well (*Figure 6K*). This phenomenon is independent of the order of hemiray removal: when the distal adjacent hemiray (segment C) was removed first, osteoblasts migrated distally (*Figure 6L*). When we then removed the proximal adjacent hemiray (segment A), the distal osteoblasts continued to migrate distally, while the proximally located osteoblast migrated towards the new injury site, even though at this site no blastema will form (*Figure 6L*). These data strongly suggest that all bone injuries release signals that attract osteoblasts and that osteoblasts are equally likely to migrate proximally and distally. Thus, the lack of regenerative growth and bone formation at proximally-facing injuries cannot be explained by an intrinsic polarity or bias in the migration of osteoblasts on the bone matrix along the proximodistal axis of the fin.

At 2 dpi, at the distal-facing injury site, *bglap*:GFP+ osteoblasts of the centre segment accumulated beyond the bone matrix in the defect, where a blastema appears to form (asterisk in *Figure 6J*). However, no osteoblasts could be observed in the defect at the proximal side of the centre segment at 3 dpi (*Figure 6J*). Thus, while osteoblasts migrate along the bone matrix towards both injury sites, they exclusively migrate off the centre segment and into the defect at the distal-facing injury. In a cavity injury model, Cao et al. have shown that inhibition of the $Ca^{2+}$/calmodulin-dependent phosphatase calcineurin can induce blastema formation at proximal-facing injuries (*Cao et al., 2021*). To analyse if such an intervention can also trigger osteoblast migration beyond the bone matrix at the proximal injury site, we treated fish with the calcineurin inhibitor FK506. Indeed, we could observe recruitment of *bglap*:GFP+ cells into the defect at the proximal site in FK506 treated fish (*Figure 6M*). Surprisingly, however, migration beyond the distal amputation site was also enhanced (*Figure 6M*). Therefore, to test whether calcineurin inhibition specifically enabled migration of osteoblasts into proximally-facing bone defects or whether it generally facilitated migration in all directions, we analysed osteoblast migration after regular fin amputation. FK506 treatment enhanced osteoblast migration at 1 dpa (*Figure 6—figure supplement 1C*), indicating that calcineurin might generally negatively regulate osteoblast migration and suggesting that it does not specifically facilitate distally directed regenerative responses.

The lack of osteoblast accumulation at proximally-facing injuries could be due to absence of chemical or mechanical cues that allow them to migrate into the defect. Alternatively, migration could be actively inhibited or osteoblasts could be eliminated at proximal injuries. One possibility would be increased osteoblast apoptosis at proximally-facing injuries. However, we could not detect any apoptotic osteoblasts at all in the first 2 days after injury (data not shown), and also at 3 dpi the number of apoptotic osteoblasts was negligible at both the proximal and distal injury sites (*Figure 6—figure supplement 1D*). Another possibility would be that osteoclasts interfere with osteoblast differentiation and bone formation at the proximal injury. To analyse if more osteoclasts are recruited to the proximal injury in the hemiray removal model, we analysed the distribution of *cathepsinK*:YFP+ cells, a marker for osteoclasts, after hemiray removal. However, osteoclasts accumulated at both wound sites by 3 dpi (*Figure 6—figure supplement 1E*).

Bone is thicker at the proximal base of the fin than at its distal end (*Mari-Beffa and Murciano, 2010*; *Pfefferli and Jaźwińska, 2015*). We thus wondered whether this structural heterogeneity along the fin ray could determine the different regenerative outcomes at proximal vs distal injuries. Yet, we did not observe a measurable difference in the diameter, radius, and thickness of one hemiray segment at its proximal vs distal end (*Figure 6—figure supplement 1F*). This suggests that structural heterogeneities of the bone along the proximodistal axis are too small at the scale of the centre segment to explain the radically different regenerative outcome at the proximal vs distal injury in the hemiray removal model.

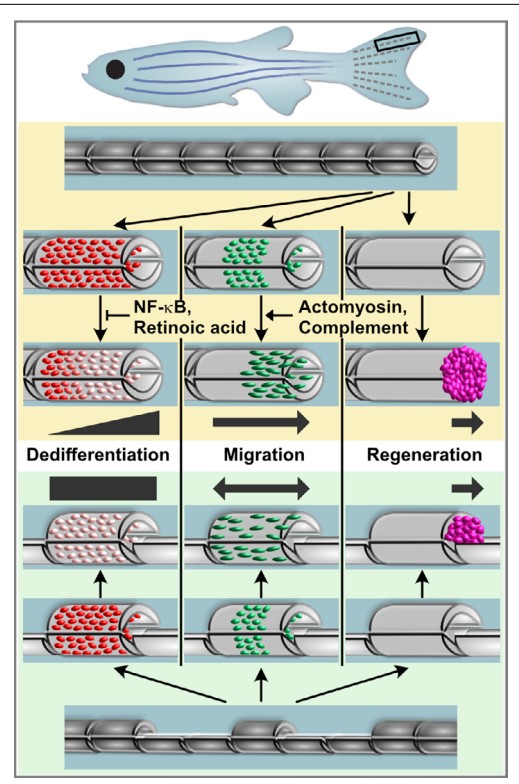

**Figure 7.** Model for osteoblast responses to fin amputation (upper panel) and hemiray removal (lower panel). Osteoblast dedifferentiation and migration represent generic injury responses that are differentially regulated and can occur independently of each other and of regenerative bone growth. Upper panel: After fin amputation, osteoblasts downregulate the expression of differentiation markers. The extent depends on their distance to the amputation site, with osteoblasts close to the amputation site displaying more pronounced dedifferentiation. In addition, osteoblasts elongate and migrate towards the amputation plane and beyond to found osteogenic cells in the blastema (pink). Thus, osteoblast dedifferentiation, migration, and bone regeneration are all distally oriented. Osteoblast dedifferentiation is negatively regulated by NF-$\kappa$B and retinoic acid signalling, while actomyosin dynamics and the complement system are required for directed osteoblast migration. Lower panel: In the hemiray removal model, a proximal and a distal injury are created on both sides of a remaining centre segment. The extent of osteoblast dedifferentiation is even along the centre segment, and osteoblasts migrate towards both injury sites. Yet, only at the distally-facing site, osteoblasts migrate into the bone defect, and blastema formation and bone regeneration only occur here.

## Discussion

### Generic and regeneration-specific responses of osteoblasts to injury

In this study, we interrogate the inter-relatedness of several in vivo responses of osteoblasts to injury using the zebrafish fin model. After fin amputation, osteoblasts elongate and orient along the proximodistal axis of the fin and migrate distally towards the amputation plane (*Figure 7*, upper panel). In addition, osteoblasts dedifferentiate, that is, they downregulate expression of differentiation markers like *bglap* and *entpd5* and initiate expression of the preosteoblast marker *runx2a*. Finally, they proliferate and migrate off the bone surface to found a population of osteogenic progenitors within the regeneration blastema that drives bone regeneration (*Figure 7*, upper panel). Surprisingly, we find that these injury responses appear to occur largely independently of each other. In particular, osteoblast dedifferentiation does not seem to be a prerequisite for osteoblast cell shape changes and migration. These conclusions are based on two lines of evidence: first, we identified molecular interventions that can interfere with one process without affecting the others: RA and NF-$\kappa$B signalling pathways regulate only osteoblast dedifferentiation, but not cell shape changes and migration, while the complement and actomyosin systems are required for osteoblast cell shape changes and migration, but not for dedifferentiation (*Figure 7*, upper panel). Second, using a hemiray injury model, where a proximally- and distally-facing bone injury is introduced in the same fin ray, we find that dedifferentiation, migration, and regenerative bone growth do not occur at both injuries. Osteoblasts migrate towards both sites and dedifferentiate equally in the proximity of both injuries (*Figure 7*, lower panel). Stunningly, however, osteoblast migration beyond the bone into the injury defect, formation of a blastema containing a preosteoblast population, and regenerative bone growth only occur at the distally-facing injury (*Figure 7*, lower panel). Thus, osteoblast migration and dedifferentiation represent generic injury responses that can occur independently of a regenerative response.

The fact that dedifferentiation and migration do not always result in bone regeneration raises questions about their function. We have previously shown that experimentally enhanced dedifferentiation results in an increased population

of progenitor cells in the regenerate, which, however, had detrimental effects on bone maturation (*Mishra et al., 2020*), indicating that dedifferentiation must be kept within certain limits. Unfortunately, currently existing tools to block dedifferentiation are either mosaic (activation of NF-$\kappa$B signalling using the Cre-lox system) or cannot be targeted to osteoblasts alone (treatment with RA). Due

to these limitations in our assays, we can currently not test what consequences specific, unmitigated perturbation of osteoblast dedifferentiation has for overall fin/bone regeneration. Conversely, the interventions presented here that specifically perturb osteoblast migration are limited as they act only transiently, that is, they can severely delay, but not fully block migration. Furthermore, while interference with actomyosin dynamics reduces regenerative growth, we cannot distinguish whether this is caused by the inhibition of osteoblast migration or due to other more direct effects on cell proliferation and tissue growth. Thus, an unequivocal test of the importance of osteoblast migration for bone regeneration requires different tools. However, it has been shown that fin bone can regenerate even after genetic ablation of osteoblasts, due to activation of non-osteoblastic cells which can drive de novo osteoblast differentiation (*Singh et al., 2012*; *Ando et al., 2017*). We speculate that the robustness of bone regeneration in zebrafish is enhanced by redundant mechanisms of source cell formation, which can either derive from pre-existing osteoblasts, which dedifferentiate and migrate towards the injury, or from (reserve) stem/progenitor cells. Dedifferentiation and migration in response to injuries that eventually do not trigger bone regeneration likely add to the robustness, since they prime all injuries for regeneration. Interestingly, work in planaria and zebrafish fins has previously shown that all wounds, irrespective of whether they trigger structural regeneration or result merely in wound healing, appear to activate a generic signalling response (*Wurtzel et al., 2015*; *Owlarn et al., 2017*). The differential outcome (structural regeneration vs wound healing) is rather determined further downstream and might dependent on systems that measure how much anatomy is missing (*Owlarn et al., 2017*). In the present work, we show that this principle also applies to injury responses that occur within osteoblasts. It appears that generic wounding-induced signals trigger the generic responses of osteoblast dedifferentiation and migration, while the decision whether these will be followed by bone regeneration is controlled by other determinants further downstream. Currently, the molecular nature of determinants that can sense the amount of missing anatomy and determine the difference between proximally and distally facing injuries remains rather mysterious.

## The complement system regulates osteoblast migration in vivo

More than 20 different factors have been found to elicit a migratory response of osteoblasts in in vitro gain-of-function assays, including the activated complement peptides C5a and C3a (*Dirckx et al., 2013*; *Thiel et al., 2018*). However, confirmation of the role of many of these factors in regulating osteoblast migration in in vivo models is still sparse, largely due to the difficulty in assaying osteoblast migration independently of other phenotypes, for example, osteoblast proliferation or differentiation, in rodents in vivo. Using live zebrafish and time lapse imaging, we show that the complement system regulates osteoblast migration in vivo. Thus, the bone fracture repair defects observed in C5-deficient mice (*Ehrnthaller et al., 2013*) might be due to impaired recruitment of osteoblasts to the injury. The *c3ar1* and *c5aR1* receptors are expressed in mature fin osteoblasts, making it likely that the complement cascade directly regulates osteoblast migration. Interestingly, our data indicate that in zebrafish, both C3a and C5a act as guidance cues for osteoblasts, ensuring the recruitment of osteoblasts to the injury plane.

There is growing evidence that complement factors can modulate highly diverse processes such as cell growth, differentiation, and regeneration in various tissues. Both C3 and C5 were shown to play a role during liver regeneration in mammals, as $C5^{-/-}$, $C3^{-/-}$ as well $C3R^{-/-}$ deficient mice show impaired liver regeneration (*Mastellos et al., 2001*; *Markiewski et al., 2004*). Expression of *c5aR1* is upregulated in regenerating hearts of zebrafish, axolotls, and neonatal mice, and receptor inhibition impairs cardiomyocyte proliferation (*Natarajan et al., 2018*). Noteworthy, in our study on zebrafish fin regeneration, inhibition of C5aR1 had no effect on osteoblast proliferation and regenerative growth. The distinct outcome of complement activation might be dependent on the location of complement factors. Intracellular C3 and C5 storage has been proposed in T cells, and the respective cleavage products C3a and C5a bind to intracellular receptors (*Liszewski et al., 2013*; *Arbore et al., 2016*). In human CD4[+] T cells, intracellular C3a regulates cell survival, while activation of membrane-bound C3aR modulates induction of the immune response (*Liszewski et al., 2013*). In addition to the receptors, both C3 and C5 are expressed in mammalian osteoblasts (*Ignatius et al., 2011b*). While we lack the tools to determine the spatiotemporal distribution of the C3 and C5 precursor proteins and their activated cleavage products C3a and C5a, our expression data indicates that the majority of the precursors are produced in the liver and are distributed via the circulation. Interestingly, however,

local, injury-induced transcription of *c5* and the *c3* paralog *c3a.5* might contribute to activation of the cascade in response to fin injury as well. Intriguingly, during newt and urodele limb regeneration, C3 is expressed in blastema cells, which implies a role in mediating regeneration (*Del Rio-Tsonis et al., 1998*; *Kimura et al., 2003*). However, as during zebrafish fin regeneration osteoblasts migrate before blastema formation, a different cellular source likely produces C3 and C5. Yet, we cannot exclude that complement peptides emanating from the blastema are responsible for the migration of osteoblasts beyond the bone context at later stages, which we only observed at sites where a blastema forms. Prior to blastema formation, the injury site is covered by a wound epidermis. During newt limb regeneration, C5 was shown to be absent from the blastema but highly expressed in the wound epidermis (*Kimura et al., 2003*). Thus, complement activation in epidermal cells might regulate the early migration of osteoblasts to the injury.

### Determinants of regeneration-specific osteoblast injury responses

Removing part of the bone within a ray results in polarised bone regeneration that is exclusively directed towards the distal end of the fin. In a recently published zebrafish fin cavity injury model, blastema markers were shown to be solely upregulated at the distal facing injury (*Cao et al., 2021*). In our hemiray-removal model, we analyse the proximal and distal injury of a centre segment. Thus, both injury sites are equally severed from innervation and blood supply, giving us confidence that the differential regenerative response at proximal and distal injuries is not due to differences in circulation or innervation. The distally oriented polarised regenerative response implies intrinsic mechanisms regulating diverse outcome from seemingly equal injuries (the removal of the adjacent hemiray). Our findings show that such regulators of distally- oriented regeneration seem to act downstream to or independently of regulators of other injury responses like osteoblast migration, dedifferentiation, and proliferation, that occur at all injuries. One previously suggested regulator of distally- oriented regeneration is the Ser/Thr phosphatase calcineurin (*Cao et al., 2021*). During zebrafish fin regeneration, inhibition of calcineurin increases regenerative growth (*Kujawski et al., 2014*). Importantly, the polarised, distally-oriented blastema formation in the cavity injury model can be overridden by calcineurin inhibition, which induces blastema formation also at the proximal injury (*Cao et al., 2021*). We also found that calcineurin inhibition can trigger the recruitment of osteoblasts into the bone defect at the proximal injury site in our hemiray removal model. Yet, we also observed increased distally-oriented osteoblast migration upon FK506 treatment, suggesting that calcineurin does not specifically regulate the polarity of injury responses along the proximodistal axis, but generally dampens responses at all injuries, including osteoblast migration and cell proliferation.

In conclusion, our findings support a model in which zebrafish fin bone regeneration involves both generic and regeneration-specific injury responses of osteoblasts. Morphology changes and directed migration towards the injury site as well as dedifferentiation represent generic responses that occur at all injuries even if they are not followed by regenerative bone formation. While migration and dedifferentiation can be uncoupled and are (at least partially) independently regulated, they appear to be triggered by signals that emanate from all bone injuries. In contrast, migration off the bone matrix into the bone defect, formation of a population of (pre-)osteoblasts, and regenerative bone formation represent regeneration-specific responses that require additional signals that are only present at distal-facing injuries. The identification of molecular determinants of the generic vs regenerative responses will be an interesting avenue for future research.

## Materials and methods

**Key resources table**

| Reagent type (species) or resource | Designation | Source or reference | Identifiers | Additional information |
|---|---|---|---|---|
| Genetic reagent (*Danio rerio*) | *bglap*:GFP; *Ola.Bglap.1*:EGFP[hu4008] | *Knopf et al., 2011* | ZDB-ALT-110713–1 | |
| Genetic reagent (*Danio rerio*) | *osx*:CreER; *OlSp7*:CreERT2-p2a-mCherry[tud8] | *Knopf et al., 2011* | ZDB-TGCONSTRCT-100928–1 | |

*Continued on next page*

*Continued*

| Reagent type (species) or resource | Designation | Source or reference | Identifiers | Additional information |
|---|---|---|---|---|
| Genetic reagent (*Danio rerio*) | *hs*:R to G; *hsp70l*:loxP DsRed2 loxP nlsEGFP[tud9] | *Knopf et al., 2011* | ZDB-TGCONSTRCT-100928–2 | |
| Genetic reagent (*Danio rerio*) | *hs*:Luc to IKKca BFP; *hsp70l*:loxP Luc-myc Stop loxP IKKca-t2a-nls-mTagBFP2-V5, cryaa:AmCyan[ulm12Tg] | *Mishra et al., 2020* | ZDB-ALT-181018–2 | |
| Genetic reagent (*Danio rerio*) | *entpd5*:kaede; TgBAC(entpd5a:Kaede) | *Huitema et al., 2012* | ZDB-TGCONSTRCT-150223–1 | |
| Genetic reagent (*Danio rerio*) | *cathepsinK*:YFP | *Apschner et al., 2014* | | |
| Genetic reagent (*Danio rerio*) | *hs*:Luc to nYPet IκBSR; *hsp70l*:loxP Luc2-myc Stop loxP nYPet-p2a-IκBSR, *cryaa*:AmCyan[ulm15Tg] | This paper | | Cre responder line to manipulate NF-κB signalling |
| Antibody | Anti-Zns5 (mouse monoclonal) | Zebrafish resource center, Eugene USA | RRID:AB_10013796 | IF (1:300) |
| Antibody | Anti-Runx (mouse monoclonal) | Santa Cruz | RRID:AB_1128251 | IF (1:300) |
| Antibody | Anti-Osterix (rabbit polyclonal) | Santa Cruz | RRID:AB_831618 | IF (1:300) |
| Commercial assay or kit | RNAscope Multiplex Fluorescent kit v2 | ACD Bio-techne | Cat# 323,100 | |
| Commercial assay or kit | ApopTag Red In Situ Apoptosis Detection Kit | Merck | Cat# S7165 | |
| Commercial assay or kit | EdU-Click 647 kit | Baseclick GmbH | Cat# BCK-EdU647 | |
| Commercial assay or kit | LunaScript RT SuperMIX Kit | NEB | Cat# E3010 | |
| Chemical compound, drug | Cytochalasin D | Calbiochem | Cat# 250,255 | |
| Chemical compound, drug | (-)-blebbistatin | Sigma | Cat# B0560 | |
| Chemical compound, drug | W54011 | Enzo | Cat# ENZ-CHM122-0001 | |
| Chemical compound, drug | SB290157 | Cayman | Cat# 15,783 | |
| Chemical compound, drug | PMX205 | Tocris | Cat# 5196 | |
| Chemical compound, drug | Retinoic acid | Sigma | Cat# R2625 | |
| Chemical compound, drug | Nocodazole | Sigma | Cat#M1404 | |
| Chemical compound, drug | FK506 | Merck | Cat# F4679 | |
| Chemical compound, drug | (Z)–4-Hydroxytamoxifen | Sigma Aldrich | Cat# H7904 | |
| Software, algorithm | Fiji | *Schindelin et al., 2012* | | |
| Software, algorithm | GraphPad Prism 9 | GraphPad software | | |

## Animals

All procedures involving animals adhered to EU directive 2010/63/EU on the protection of animals used for scientific purposes and were approved by the state of Baden-Württemberg (Project numbers 1193 and 1494) and by local animal experiment committees. Fish of both sexes were used. Housing

and husbandry followed the recommendations of the Federation of European Laboratory Animal Science Associations (FELASA) and the European Society for Fish Models in Biology and Medicine (EUFishBioMed) (*Aleström et al., 2020*).

The following pre-existing transgenic lines were used: *bglap*:GFP (*Ola.Bglap.1*:EGFP[hu4008]; *Knopf et al., 2011*), *osx*:CreER (*Ola.Sp7*:CreERT2-p2a-mCherry[tud8]; *Knopf et al., 2011*), *entpd5*:kaede (TgBAC[entpd5a:Kaede]; *Huitema et al., 2012*), *cathepsinK*:YFP (*Apschner et al., 2014*), *hs*:R to G (*hsp70l*:loxP DsRed2 loxP nlsEGFP[tud9]; *Knopf et al., 2011*), *hs*:Luc to IKKca BFP (*hsp70l*:loxP Luc-myc Stop loxP IKKca-t2a-nls-mTagBFP2-V5, *cryaa*:AmCyan[ulm12Tg]; *Mishra et al., 2020*). To create the line *hsp70l*:loxP Luc2-myc Stop loxP nYPet-p2a-IκBSR, *cryaa*:AmCyan[ulm15Tg], in short *hs*:Luc to nYPet IκBSR, the following elements were assembled by Gibson assembly and restriction-based cloning methods: MiniTol2 inverted repeat, attP site, zebrafish *hsp70l* promoter, loxP, firefly luciferase 2, 6× myc tag, ocean pout antifreeze protein polyA signal, loxP, nYPet, p2a, IκBSR (IkappaB super-repressor), SV40 polyA signal, zebrafish *cryaa* promoter, AmCyan, SV40 polyA signal, MiniTol2 inverted repeat. IkBSR is a mutated version of mouse IKappa B alpha (Entrez gene, Nfkbia A1462015), described in *Van Antwerp et al., 1996*, whose inhibitory N-terminal Serine residues 32 and 36 are mutated to Alanine. Phosphorylation sites in the C-terminal PEST domain are also mutated: Serines 283, 288, and 293 are converted to Alanine; Threonines 291, 296 also to Alanine, and Tyrosine 302 to Aspartic acid. Tol2-mediated transgene insertion was used to create a stable transgenic line. One subline was selected based on widespread expression after heat shock in the adult fin and efficient recombination in embryos when crossed with a ubiquitin promoter-driven Cre driver line (unpublished).

To achieve optimal expression levels of the *osx*:CreER transgene, we activated its expression in adult fins as described previously (*Mishra et al., 2020*). Briefly, fish were amputated and allowed to regenerate for 8 days, at which timepoint mineralised bone has formed in the regenerate. We then reamputated through the mineralised part of the regenerate and analysed the stump of this second amputation.

## Fin amputations and hemiray removal

Adult zebrafish were anaesthetised with 625 µM tricaine, and the caudal fins were amputated through the second segment proximal to the bifurcation. For hemiray removal, small surgical blades were used to cut through four joints proximal to the bifurcation in the second ray from ventral and dorsal, and fine tweezers were used to remove the upper, left hemiray (facing towards the experimenter) of the segments located proximal and distal of one central segment. For analysis of migration and dedifferentiation, one hemiray segment was removed on either side. For analysis of proliferation and Runx2 and Osterix expression, two hemiray segments were removed on both sides. For alizarin red staining at 5 dpi, three hemiray segments were removed. Fish were allowed to regenerate at 27–28.5°C.

## qRT-PCR

Total RNA was extracted from fins using RNeasy Mini Kit (Qiagen, 74104). For liver RNA, six adult livers were pooled for each replica, for non-injured fins, four whole fins were pooled, and for 6 hpa fin samples, segment –1 and segment 0 from 15 fins were pooled. cDNA synthesis was carried out using LunaScript RT SuperMIX Kit (New England Biolabs, E3010L) with random primers. Quantitative real time PCR was performed using Luna Universal qPCR Master Mix (NEB, M 3003 L) with the following primers:*c5* (ZDB-GENE-120510–2, fw: CCGGTCACTACAGTCAATGGT, rv: CGTTGAGGGAGTAAACACGC), *c5ar* (ZDB_GENE-190430–1, fw: TCGGATGCCAAACTCAGTGA, rv: CATGAGGATGGGAAGGGACA), *c3a.1* (ZDB-GENE-990415–35, fw: GGAGATGGACCAGAGTGTGT, rv: ATCAATCTCCTCCCACAGCC), *c3a.2* (ZDB-GENE-990415–36, fw: CGGTACACAAACACCCCTCT, rv: GTCTTCCTCGTCGTTCTCTTGTT), *c3a.3* (ZDB-GENE-990415–37, fw: TGTTGATGAATCTAAGCGCTTGA, rv: CGGAGTCTTGTTTCAGGTCC), *c3a.4* (ZDB-GENE-140822–3, fw: TGGTGGCTGTGGATAAAGGT, rv: CTGTGCAGCCAGTGTCATG), *c3a.5* (ZDB-GENE-090311–30, fw: ACCAAGAATCTCTGACTGTGGA, rv: CCGCCATGCTGATCTTCTTC), *c3a.6* (ZDB-GENE-041212–2, fw: TCTGGAAGGTGGTCACAAGA, rv: TCGATGCTAACCGTCAGACT), and *c3aR1* (ENSDARG00000031749, fw: CATGCTGGCTGTTATCGTGG, rv: AACACATACAGGACGGGGTT). Relative quantification was performed using the ΔΔCt method, and the Ct values of each gene were normalised to the arithmetic mean of two housekeeping genes, *actb2* (ZDB-GENE-000329–3, fw: ACGATGGATGGGAAGACA, rv: AAATTGCCGCACTGGTT) and *hatn10* (ZDB-NUCMO-180807–7, fw: TGAAGACAGCAGAAGTCAATG, rv: CAGTAAACATGTCAGG

CTAAATAA). Each biological sample was run in technical duplicates and results were averaged. qPCRs and statistics were performed on two biological replicates for liver samples, three biological replicates for non-injured fin samples, three biological replicates for injured fin samples, and four biological replicates for cells isolated by FACS (Fluorescence Activated Cell Sorting).

## Fluorescence Activated Cell Sorting of osteoblasts

Differentiated osteoblasts were isolated at 1 dpa from segment 0 and segment –1 of adult *bglap*:GFP transgenics using FACS as described by *Lee et al., 2020*. A total of 15 fins were pooled for each of the 4 replicates. In brief, samples were treated with 1× TrypLE Express enzyme (Thermo Fisher, 12604013) at 37°C for 30 min with gentle agitation, washed with cold Dulbecco's phosphate buffered saline (DPBS) buffer (Thermo Fisher, 14190136), and further dissociated into a single cell suspension with 0.25 mg/mL Liberase DL (Merck, 5401160001) in enzyme-free Cell Dissociation Buffer (Thermo Fisher, 13151014) at 37°C for 30 min with gentle agitation. The dissociated single cell suspension was washed with cold DPBS buffer and pelleted by centrifugation at 500× g for 3 min at 4°C. Cells were resuspended in cold DPBS buffer with 2% fetal bovine serum and filtered through a 70 µm sample preparation filter (pluriSelect, 43-10070-46) to remove cell aggregates. The FACS Aria II flow cytometer was used to separate GFP+ and GFP– populations from single-cell suspensions. In order to define GFP background levels, wildtype non-transgenic fish were used. Forward and side scatter parameters were applied to exclude debris and doublets. Dead cells were excluded by staining with the cell-impermeable DNA-dye 4',6-diamidino-2-phenylindole dihydrochloride (DAPI, Sigma, 32670) and sorting for DAPI-negative cells. To further discriminate between GFP+ anuclear cell debris and intact cells, we stained with the cell-permeable DNA dye DRAQ5 (Cell Signalling, 4084 S) and sorted for DRAQ5+ cells. Therefore, DAPI–/DRAQ5+/GFP+ cells were sorted, and the same number of DAPI–/DRAQ5+/GFP– cells from each sample were also sorted using a random GFP gate to serve as control group.

## Pharmacological interventions

10 µl of the following drugs (in PBS) were injected IP using a 100 µl Nanofil syringe (World Precision Instruments NANOFIL-100): blebbistatin 750 nM, cytochalasin D 30 µM, W54011 10 µM, SB290157 10 µM, PMX205 10 µM. Fish were injected once daily, starting 1 hr prior to injury. For the following treatments, fish were immersed in fish system water containing the following drugs: RA 5 µM, nocodazole 5 µM, FK506 3 µM. Fish were set out in drug solution 1 hr prior to injury, and solutions were changed daily. For the entire duration of the experiment, fish were kept in an incubator at 25°C in the dark at 1 fish/100 ml density in fish system water. Negative control groups were injected or soaked, respectively, with the corresponding DMSO concentration in PBS or fish system water.

## Whole mount immunohistochemistry

Fins were fixed overnight with 4% PFA at 4°C. After 2 × 5 min washes with PBTx (1× PBS with 0.5% TritonX 100) at RT, fins were transferred to 100% acetone, rinsed once, and kept for 3 hr at −20°C. Fins were transferred to PBTx, washed 2 × 5 min, 2 × 15 min, and incubated for 30 min at RT, followed by blocking in 1% BSA in PBTx for at least 1 hr at RT. Primary antibodies were diluted to 1:300 in blocking solution and fins were incubated overnight at 4°C. The next day, fins were washed several times with PBTx and incubated with secondary antibodies at 1:300 dilution in PBTx overnight at 4°C. Fins were mounted in Vectashield (Vectorlabs, H-1000). For anti-Runx2 staining, fins were fixed with 80% methanol/20% DMSO overnight at 4°C and rehydrated by a graded series of methanol in PBTx (75, 50, 25%) for 5 min each. Fins were washed in PBTx for 30 min, after which fins were processed following the above-mentioned protocol. Primary antibodies used in this study were: mouse anti-Zns5 (Zebrafish International Resource Center, Eugene, OR, USA, RRID:AB_10013796), mouse anti-Runx (Santa Cruz sc-101145, RRID:AB_1128251), and rabbit anti-Osterix (Santa Cruz sc-22536-R, RRID:AB_831618).

## RNAscope whole mount in situ hybridisation

For detection of RNA transcripts in whole mount fins, the RNAscope Multiplex Fluorescent Reagent Kit v2 (ACD Bio-techne, 323100) was used with the following probes: *bglap*-C1 (ACD Bio-techne, 519671), *cyp26b1* (ACD Bio-techne, 571281-C2), *entpd5* (ACD Bio-techne, 820491-C4), *runx2a* (ACD Bio-techne, 409521-C2), and *c5aR1* (ACD Bio-techne, 859561-C2). Fins were fixed overnight in 4%

PFA at 4°C. The next day, fins were washed with PBT (1× PBS with 0.1% Tween) 3 × 5 min each. After incubation in RNAscope hydrogen peroxide for 10 min, fins were rinsed 3× with PBT, treated with RNAscope protease plus for 20 min, and rinsed 3× with PBT. All these steps were performed at RT. Probes and probe diluent were prewarmed to 40°C and cooled down to RT before use. The fins were incubated with probes overnight at 40°C. The next day, fins were washed 3× with 0.2× SSCT, 12 min each at RT. All the steps mentioned from here on were followed by such washing steps. Fins were post fixed with 4% PFA for 10 min at RT and incubated with AMP1, AMP2, and AMP3 for 30, 15, and 30 min, respectively, at 40°C. To develop the signal, fins were incubated with the corresponding RNAscope Multiplex FL v2 HRP for 15 min at 40°C. TSA-fluorophore (Perkin Elmer NEL701A001KT) was used in a 1:1500 dilution in TSA buffer (RNAscope kit). Fins were incubated for 15 min at 40°C.

## Proliferation assay

To analyse cell proliferation, 5-ethynyl-2′-deoxyuridine (EdU)-Click 647 kit (baseclick GmbH BCK-EdU647) was used. Fish were IP injected with 10 µl of 10 mM EdU in PBS using a 100 µl Nanofil syringe (World Precision Instruments NANOFIL-100). In amputation assays, fish were injected as indicated in the figures. In the hemiray removal assay, fish were injected once at 1 dpi for evaluation at 2 dpi, and at 1 and 2 dpi for evaluation at 3 dpi. Fins were fixed with 4% PFA overnight and EdU labelling was performed according to the manufacturer's protocol. If applicable, EdU labelling was followed with AB staining following the immunohistochemistry protocol, or RNAscope in situ following the hybridisation protocol.

## Apoptosis assay

To analyse apoptosis, the ApopTag Red In Situ Apoptosis Detection Kit (Merck, S7165) was used. Fins were fixed overnight in 80% Methanol/20% DMSO at 4°C. The next day, fins were rehydrated through a Methanol/PBTx (PBS +0.5% TritonX 100) series (75, 50, 25%; 5 min each) and permeabilised with acetone for 3 hr at –20°C. Fins were transferred to PBTx, washed 2 × 5 min, and incubated for 30 min at RT, followed by blocking in 1% BSA in PBTx for at least 1 hr at RT. Mouse anti-Zns5 (Zebrafish International Resource Center, Eugene, OR, USA, RRID:AB_10013796) was diluted to 1:300 in blocking solution and fins were incubated overnight at 4°C. The next day, fins were washed several times with PBTx and transferred to superfrost slides. Using a hydrophobic pen, a circle was drawn around the fins. Fins were first equilibrated with 75 µl equilibration buffer (ApopTag kit) for 10 s at RT, followed by incubation in 55 µl TdT enzyme (ApopTag kit) in a humidified chamber at 37°C for 1 hr. The reaction was stopped by several washes with Stop/Wash buffer (ApopTag kit), followed by an incubation in Stop/Wash buffer for 10 min at RT. Fins were washed 3 × 1 min with PBS and incubated in anti-DIG-Rhodamine diluted 1:2000 in blocking solution (ApopTag kit) and secondary antibody (1:300) overnight at 4°C. The next day, fins were washed 6 × 20 min at RT and mounted with Vectashield (Vectorlabs, H-1000).

## Alizarin red S staining

In vivo staining with alizarin red S (ARS; Sigma-Aldrich A3757) was performed as previously described (*Bensimon-Brito et al., 2016*). Briefly, fish were stained in 0.01% ARS in system water for 15 min at RT, washed 3 × 5 min in system water, immediately anaesthetised, and imaged.

## Cre-lox recombination and heat shocks

Adult double transgenic fish (*OlSp7*:CreERT2-p2a-mCherry[tud8]; *hsp70l*:loxP DsRed2 loxP nlsEGFP[tud9], *OlSp7*:CreERT2-p2a-mCherry[tud8]; *hsp70l*:loxP Luc2-myc Stop loxP nYPet-p2a-IκBSR, *cryaa*:AmCyan[ulm15Tg] and *OlSp7*:CreERT2-p2a-mCherry[tud8]; *hsp70l*:loxP Luc-myc Stop loxP IKKca-t2a-nls-mTagBFP2-V5, *cryaa*:AmCyan[ulm12Tg]) were IP injected with 10 µl of 3.4 mM 4-OH tamoxifen (4-OHT) in 25% ethanol once daily for 4 days. Fish were heat shocked twice (at 24 hr and 3 hr prior to harvest) at 37°C for 1 hr after which water temperature was returned to 27°C within 15 min.

## Imaging

Images in *Figures 1F, G, 3C, G and 6J*, *Figure 5E*; *Figure 5—figure supplement 4A, B* and *Figure 6—figure supplement 1E* were acquired with a Leica M205FA stereo microscope and display live fluorescence of fluorescent proteins. High-resolution optical sections were obtained with a Leica SP8

confocal microscope or a Zeiss AxioObserver 7 equipped with an Apotome and processed using Fiji (*Schindelin et al., 2012*). Images in *Figures 1B,C and D, 2A, D, 4A and 6B,E,G*; and *Figure 1—figure supplement 1A, B*; *Figure 2—figure supplement Figure 2—figure supplements 1A and 2A,B*, *Figure 5—figure supplement 1A*; *Figure 6—figure supplement 1B,D* and *Video 1* are z-projections. *Figure 5A* and *Figure 5—figure supplement 2D* show single z-planes. *Video 2* was recorded with a Leica M205FA stereo microscope. Movie annotations were added using the Annotate_movie plugin (*Daetwyler et al., 2020*).

## Cell morphology quantification

To quantify osteoblast cell shape and orientation, the transgenic line *bglap*:GFP in combination with Zns5 AB labelling was used. Osteoblasts of the outer layer of one hemiray (facing the objective in whole fin mounting) were imaged and analysed. As Zns5 localises to the plasma membrane of all osteoblasts, the combination of both markers provides solid definition of single cell outlines. All GFP+ Zns5+ cells with such a defined outline within an analysed segment were included into the analysis, and cells along the whole proximodistal axis of a segment were measured. In the transgenic intervention studies, mCherry is expressed under the *osx* promoter and was used as cytosolic labelling of osteoblasts. Using Fiji (*Schindelin et al., 2012*), the longest axis of a FP+ Zns5+ cell was measured as maximum length, the short axis as maximum width, and the ratio calculated. Simultaneously, the angle of the maximum length towards the proximodistal ray axis was measured for angular deviation. All measurements were performed manually, with the analyst being blinded.

## Bulk migration assay

To quantify osteoblast migration, fins of *bglap*:GFP transgenic fish were imaged with a GFP filter and in brightfield at 0 and 1 dpa with a Leica Stereomicroscope M205FA. Using Fiji (*Schindelin et al., 2012*), a threshold was set for the GFP signal to exclude background fluorescence and to select the bulk of GFP+ cells. The thresholded image was merged with the corresponding brightfield image and the distance between GFP+ cells and the joint at the ventral-distal centre of the segment was measured. For each fin, the second and third rays in the dorsal lobe of the fin were analysed. Statistical analysis was performed on the difference in distance between 0 and 1 dpa, while for illustration the change at 1 dpa is plotted, with migration across the whole distance to the joint counting as 100% migration.

## RNAscope quantification

For quantification of RNAscope signals, optical sections were acquired with a Zeiss AxioObserver 7 microscope equipped with an Apotome, using identical imaging settings within one experiment. To analyse *bglap* intensity along the proximodistal axis, subsequent ROIs (segment height × 50 µm width) were analysed (joints excluded), and intensity was normalised for each ray. For spatial resolution of single osteoblast intensity, segment lengths were normalised and x-location of osteoblasts grouped into proximal (0.0–0.2 normalised segment length), central (0.4–0.6), and distal (0.8–1.0) regions. For categorising *bglap* expression of single cells, osteoblasts were grouped into three classes based on *bglap* expression intensity using a look-up table (LUT). LUT was designed so that the 'high' threshold (pixel values 192–255) corresponds to the expression levels in segment –2 and to further robustly discriminate between 'low' (32-127) and 'medium' (128-191) cells.

## Regenerative growth quantification

Regenerative growth was analysed in brightfield images taken at 3 dpa. For each fin, the length of the second and third dorsal ray regenerate was measured from the amputation plane to the distal tip and the average calculated. For testing significance of growth differences between drug treatments and control fish, the data were box cox transformed using the formula $\frac{x^{\lambda}-1}{\lambda}$ with $\lambda$ =1.5, which we determined previously (*Mishra et al., 2020*).

## Statistical analysis

Fish were randomly allocated into experimental groups. Statistical analyses were performed using GraphPad Prism (GraphPad software, LCC). For proliferation analyses in the hemiray removal model, a random distribution set was generated with MS Excel (Microsoft Corporation). Percentage data were

arcsin transformed for statistical analysis. For experiments where we did not observe significant differences between experimental groups, if applicable we used the following logic to support statements that no differences exist or that these are not biologically relevant: the smallest difference that would have been significant was calculated based on the effect size and the SDs and sample sizes of the experimental groups. Using G × Power (*Faul et al., 2009*) the effect size was computed with these parameters: $t$-test, tails=2, $\alpha$=0.05, power $(1-\beta)$=0.8, and the sample size of both groups. For non-normal distributed data sets, data was log-transformed. These calculated smallest significant differences and the observed differences are reported in the figure legends. If the calculated smallest significant difference is smaller than significant differences we observed with similar assays in other experiments, we conclude that the experiment would have had enough statistical power to detect a similar effect size.

## Acknowledgements

We thank Doris Weber and Marion Gradl for their contributions to fish care and the core facility 'Confocal and multiphoton imaging' of the Medical faculty of Ulm University for help with imaging. This work was funded by the Deutsche Forschungsgemeinschaft (DFG, German Research Foundation) Project-ID 251293561 – SFB 1149, project ID 316249678 – SFB 1279, and project ID 450627322 – SFB 1506. Ivonne Sehring was funded by the Hertha-Nathorff-Program of the Medical Faculty of Ulm University.

## Additional information

### Funding

| Funder | Grant reference number | Author |
| --- | --- | --- |
| Deutsche Forschungsgemeinschaft | Project-ID 251293561 - SFB 1149 | Gilbert Weidinger |
| Deutsche Forschungsgemeinschaft | project ID 316249678 - SFB 1279 | Gilbert Weidinger |
| Deutsche Forschungsgemeinschaft | project ID 450627322 - SFB 1506 | Gilbert Weidinger |
| Medical Faculty, Ulm University | Hertha-Nathorff-Program | Ivonne Sehring |

The funders had no role in study design, data collection and interpretation, or the decision to submit the work for publication.

### Author contributions

Ivonne Sehring, Conceptualization, Formal analysis, Funding acquisition, Investigation, Methodology, Validation, Visualization, Writing – original draft; Hossein Falah Mohammadi, Formal analysis, Investigation; Melanie Haffner-Luntzer, Anita Ignatius, Markus Huber-Lang, Resources, Writing - review and editing; Gilbert Weidinger, Conceptualization, Funding acquisition, Methodology, Project administration, Resources, Supervision, Writing – original draft

### Author ORCIDs

Ivonne Sehring ![ORCID] http://orcid.org/0000-0002-7812-0278
Anita Ignatius ![ORCID] http://orcid.org/0000-0002-4782-1979
Gilbert Weidinger ![ORCID] http://orcid.org/0000-0003-3599-6760

### Ethics

All procedures involving animals adhered to EU directive 2010/63/EU on the protection of animals used for scientific purposes, and were approved by the state of Baden-Württemberg (Project numbers 1193 and 1494) and by local animal experiment committees. Fish of both sexes were used. Housing and husbandry followed the recommendations of the Federation of European Laboratory Animal Science Associations (FELASA) and the European Society for Fish Models in Biology and Medicine (EUFishBioMed) (Aleström et al., 2020).

Decision letter and Author response
Decision letter https://doi.org/10.7554/eLife.77614.sa1
Author response https://doi.org/10.7554/eLife.77614.sa2

## Additional files

### Supplementary files
• Transparent reporting form

### Data availability
All data generated or analysed during this study are included in the manuscript and supporting file.

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
