## [Editor Report]

This work is of interest to readers in the field of bone regeneration, and more broadly to readers in the field of tissue repair and regenerative medicine. The authors took advantage of a well–established in vivo model, live imaging, pharmacological inhibition, and genetic strategies to dissect the interrelations of key cellular events in zebrafish fin regeneration. The finding of how distinct generic injury responses are differentially regulated and are functioning independently from each other, is a valuable piece of information for the community.

---

## [Decision Letter]

**Decision letter after peer review:**

Thank you for submitting your article "Zebrafish fin regeneration requires generic and regeneration–specific responses of osteoblasts to trauma" for consideration by *eLife*. Your article has been reviewed by 3 peer reviewers, including Céline Colnot as the Reviewing Editor and Reviewer #1, and the evaluation has been overseen by Didier Stainier as the Senior Editor.

Essential revisions:

1) Additional experimental details should be provided to demonstrate the absence of biased selection of cells chosen to be incorporated into the statistical analyses. A clearer picture of how the authors obtained their final numbers starting from the original image data is needed.

2) Additional sets of marker genes (in addition to bglap:GFP downregulation) and long time–lapse imaging of the reporter (preferably with histone–GFP) would be helpful to further substantiate the claims on "dedifferentiation" and "migration" of osteoblast populations. Additional characterization of bglap:GFP+ cells would be also helpful to determine if these cells also include osteoblast precursor cells.

3) Additional results are needed to support the claim that the complement system is required for injury–induced directed osteoblast migration including expression analyses of the central complement components C3 and C5 at the amputation site and showing whether c5ar1 and c3ar are solely expressed in osteoblasts, or rather broadly within tissue lining the hemirays.

4) It is unclear whether each step of cell elongation and alignment, cell migration, cell dedifferentiation and regenerative response, is required for fin regeneration following amputation. In the absence of direct evidence for the requirement of migration or dedifferentiation for the overall success of fin regeneration, the limitations should be more clearly stated, and the title modified.

5) The word trauma should be clarified as the authors do not provide evidence for a non–healing phenotype in their trauma model.

6) The conclusions referring to the relevance of the work to bone regeneration should be toned down and more focused on fin regeneration.

7) The manuscript is overall well written but could be improved to be less technical to help the readers, and the discussion should be reduced in length.

*Reviewer #1 (Recommendations for the authors):*

To determine whether each step of cell elongation and alignment, cell migration, cell dedifferentiation and regenerative response, is required for fin regeneration following amputation the authors discuss p14 that other tools will be needed. However, later time points could be assessed in this study, not only to show the long–term effects of the treatment on healing but also to show how healing eventually resolves in the two models used here. Did the authors assess the long–term effects of inhibitor treatments on the regeneration process? Does blocking the complement system or actomyosin prevent regeneration? Are both generic (migration, dedifferentiation) and regenerative growth required for regeneration? Since there are limitations in the approaches used, the authors should modify the title and replace the word "requires". The use of the term "require" should be reconsidered also in the abstract and throughout the manuscript if the requirement for healing or a specific process is not demonstrated.

Can the authors provide evidence of the source of C3 and C5, the central components of the complement system during fin regeneration? Figure 4 shows RNAscope in situ detection of c5aR1. Analysis of c3 and c5 using the same technique should be included.

Regarding the trauma model, it is unclear whether the gaps on each side of the remaining segment eventually heal. Does the gap on the proximal side of the remaining segment heal via regeneration occurring on the distal end of the next segment? The term "trauma" should be replaced by another term. Further, the term "trauma" appears to be used in the first paragraph of the discussion when referring to amputation and in other parts of the discussion when referring to bone injury.

The Discussion section is very long and there are many redundancies with the Results section. The main findings and conclusions could be stated more briefly, and the discussion points presented in a more concise way. As indicated in the public review, the last paragraph of the discussion should be focused on "fin regeneration" and avoid concluding on "bone regeneration" and "bone formation".

*Reviewer #2 (Recommendations for the authors):*

– Page 4, line 6: I am not sure if "exoskeletal" is by definition the right term in this context.

– What is the p–value in Figure 2A (left) between the width/length ratio of cytochalasin–treated osteoblasts from segments –1 and –2? It looks as if the cells of segment –1 are somewhat elongated even though they have been exposed to the drug. Could the authors please comment on this?

– The red labeling of data points for the drug–treated conditions in Figure 2A, B and C might be difficult to be detected by color–blind readers. This should be adjusted accordingly.

– An image showing cyp26b1 RNAscope in situ hybridization on which the graphs of Figure 5H are based should be shown somewhere – maybe in the dedicated Excel sheet of raw data.

– Suggestion for a further experiment (not required for acceptance from my side!!!): I think it would be nice to learn whether C3a and C5a are sufficient to drive directed osteoblast migration along bony elements. To test that one could try to implant a bead loaded with recombinant C3a or C5a somewhere along a fin ray to see whether this ectopic signaling source attracts osteoblasts. Would this be doable? Recombinant complement proteins appear to be commercially available.

*Reviewer #3 (Recommendations for the authors):*

1. Title:

– I feel that the current title does not capture the essence of the authors' major findings.

2. Abstract:

– The abstract is oversimplified and overstated, omitting important details. The authors should clearly state that they used a bglap:GFP strain to support their conclusions on osteoblast differentiation and migration.

3. Introduction:

– The authors make significant statements on osteoblast dedifferentiation. This is based on a naïve assumption that bglap:GFP is solely active in mature osteoblasts. However, it is clear from the authors' data that a substantial fraction of GFP+ cells are in proliferation (Figure 4 S1B). It seems that bglap:GFP is also expressed by osteoblast precursors. Specificity of bglap:GFP to mature osteoblasts is a major concern.

– Page 5, Line 5: Osteoblasts do not migrate to the resorbed sites. Osteoblast precursors (or pre–osteoblasts) do. Thiel 2018 review article does not appear to be particularly supportive.

4. Results:

– Page 6, Line 14: Perhaps the authors should explain in more detail about the kaedeGreen to Red system, and why they needed this approach.

– Page 7, Line 13–16: I do not understand why the authors interpret these data as directed active migration behavior towards the injury. What they showed here is merely cell shape changes.

– Page 7, Line 19: This data could be alternatively interpreted as the upregulation of bglap:GFP by invading cells (not previously bglap+), rather than the invasion of bglap:GFP+ cells. It would be essential to determine the half–life of GFP to assume cell invasion. The authors would need a more long–lasting histone–bound GFP reporter strain to conclusively define cell invasion.

– Page 8, Line 3–16: This paragraph is very difficult to read because of long sentences. They should simplify and rephrase the sentences.

– Page 9, Line 1: The authors should move the bglap RNAscope in situ data to the main figure, as this is an important piece of information.

– Page 9, Line 6: I don't understand why reduced bglap mRNA can be interpreted as osteoblast dedifferentiation.

– Overall, the authors would need live–cell imaging and tracking to make these conclusions.

– Page 10, Line 7: If the authors think that a bglap:GFP strain can be used as live reporters for osteoblast dedifferentiation, that would significantly compromise the scientific rigor of the study, as it would not support any conclusions regarding cell migration and invasion.

– Page 10, Line 20: Osteoblast migration should not be mixed up with cell shape changes.

– Page 11, Line 12: The authors did not show data for osteoblast differentiation/dedifferentiation in their genetic intervention studies.

– Page 11, Line 19: An alternative explanation is that retinoic acid and NF–kB signaling simply promote osteoblast differentiation, without anything to do with dedifferentiation.

– Page 12, Line 1–2: These conclusions are stretched and not supported by experimental data. The argument for the decoupling of dedifferentiation and migration is not well developed.

– Page 13, Line 8: As many as 20% of bglap:GFP+ cells are EdU+ therefore in proliferation. How would the authors explain this?

5. Discussion:

– The authors should clearly indicate their definition of osteoblasts in this study (i.e. bglap:GFP+ cells).

– Osteoblast dedifferentiation should be rephrased as reduced bglap expression.

– Page 18, Line 19: Osteoblast differentiation, instead of osteoblast dedifferentiation.

---

## [Author Response]

Essential revisions:1) Additional experimental details should be provided to demonstrate the absence of biased selection of cells chosen to be incorporated into the statistical analyses. A clearer picture of how the authors obtained their final numbers starting from the original image data is needed.

Osteoblasts line the bony hemirays on the inner and outer surface (see Figure 1A), and for quantifications of osteoblast morphology, we analysed the osteoblasts of the outer layer of one hemiray (the hemiray facing the objective in whole mount imaging). For measurement of morphology, we used a transgenic line expressing a fluorescent protein (FP) in osteoblasts in combination with Zns5 antibody labelling. Zns5 is a pan-osteoblastic marker which localizes to the cell membrane. Therefore, combination of a cytosolic FP labelling with the membrane labelling by Zns5 provides solid definition of single cell outlines. For general morphology studies and drug intervention studies, we used *bglap*:GFP transgenics. In the transgenic intervention studies (manipulation of NF-κB signalling), mCherry is expressed together with CreERT2 under the *osterix* promoter and used as cytosolic labelling of osteoblasts. Our analyses are always based on segments, e.g. we present data for segments 0, -1, 2. Within these segments all FP+ Zns5+ cells were included into the analysis, and cells along the whole proximodistal axis of a segment were measured. Measurements were performed manually, and the analysist was blinded. With these set-ups, not only a fraction but all FP+ Zns5+ osteoblasts present in those segments that we analysed were included into the analysis, and thus no selection was necessary that could have introduced bias. As suggested by Reviewer #2, we have added representative sample images to the accompanying Excel sheets of raw data for the dedicated experiments. Within these, the axes along which the measurements have been performed are indicated. We have expanded the description of the analysis in the method section. It now reads on page 36 as follows:

“To quantify osteoblast cell shape and orientation, the transgenic line bglap:GFP in combination with Zns5 AB labelling was used. Osteoblasts of the outer layer of one hemiray (facing the objective in whole fin mounting) were imaged and analysed. As Zns5 localizes to the plasma membrane of all osteoblasts, the combination of both markers provides solid definition of single cell outlines. All GFP+ Zns5+ cells with such a defined outline within an analysed segment were included into the analysis, and cells along the whole proximodistal axis of a segment were measured. In the transgenic intervention studies, mCherry is expressed under the osx promoter and was used as cytosolic labelling of osteoblasts. Using Fiji (Schindelin et al., 2012), the longest axis of a FP+ Zns5+ cell was measured as maximum length, the short axis as maximum width, and the ratio calculated. Simultaneously, the angle of the maximum length towards the proximodistal ray axis was measured for angular deviation. All measurements were performed manually, with the analyst being blinded.”

2) Additional sets of marker genes (in addition to bglap:GFP downregulation) and long time–lapse imaging of the reporter (preferably with histone–GFP) would be helpful to further substantiate the claims on "dedifferentiation" and "migration" of osteoblast populations. Additional characterization of bglap:GFP+ cells would be also helpful to determine if these cells also include osteoblast precursor cells.

To address these comments, we have performed several additional experiments as described below. In addition, we would like to refer the reviewers to our previous papers, where we have analysed the process of osteoblast dedifferentiation (Knopf et al., Dev Cell 2011, doi: 10.1016/j.devcel.2011.04.014; Geurtzen et al., Development 2014, doi: 10.1242/dev.105817; Mishra et al. Dev Cell 2020, doi: 10.1016/j.devcel.2019.11.016). Using transgenic reporters and immunofluorescence we have shown in these previous papers that osteoblasts in the non-injured fin express the differentiation marker Bglap but not the pre-osteoblast marker Runx2 (and are thus by our definition differentiated), and that these cells do not only downregulate Bglap but also upregulate Runx2 in the process of dedifferentiation.

However, we have refined and expanded our methods and are now able to determine the expression patterns of markers of the osteoblast differentiation status with single cell resolution using RNAScope in situ hybridization. We think that these data are even more conclusive than those we previously published, since they don’t rely on transgenic reporters, which might not fully reflect the endogenous expression, and are not affected by issues of antibody specificity. The results that we have now obtained with this method fully support our previous conclusions.

Already in the previous version of the manuscript we have shown that endogenous *bglap* is strongly expressed in segment -2, (the segment that does not respond to fin amputation and thus represents the non-injured state), while it is downregulated in a graded manner in segment -1 and segment 0 (the segments where dedifferentiation happens). These data were possibly somewhat hidden in the Supplement, we have now moved them to the re-designed Figure 1B. In addition to *bglap*, we can now show that *entpd5,* a gene required for bone mineralization, is strongly expressed in osteoblasts of segment -2, while it is massively downregulated in segment -1 and segment 0. Thus, *entpd5* is another differentiation marker whose loss characterizes osteoblast dedifferentiation. These new data can be found in Figure 1C. Importantly, we can confirm by RNAScope that the pre-osteoblast marker *runx2a* is absent in mature segments but is upregulated in segment 0 and segment -1 at 1 dpa (new data in Figure 1 —figure supplement 1A). Similarly, *cyp26b1*, an enzyme shown to regulate dedifferentiation, is upregulated in segment 0 and segment -1, but is not expressed in segment -2 (new data in Figure 1 —figure supplement 1B). Furthermore, we have repeated all experiments where we have previously quantified dedifferentiation upon experimental interventions using downregulation of *bglap*:GFP (actomyosin inhibition, retinoic acid treatment, complement inhibition). We now can fully confirm the previous conclusions using the more rigorous quantification of dedifferentiation using RNAScope analysis of endogenous *bglap* levels. We have replaced all *bglap*:GFP data with the new *bglap* RNAScope data. These new data are found in Figure 3F, Figure 3 —figure supplement 1A, Figure 4B and Figure 5F.

To further support our conclusion that osteoblasts do not only dedifferentiate, but also migrate, we performed time-lapse imaging using a transgenic line expressing the photoconvertible protein kaede in osteoblasts (*entpd5*:kaede). Local photoconversion of only the proximal half of a segment allowed us to trace these photoconverted osteoblasts. This revealed that converted cells appear in the distal part of the segment within 1 dpa, which can only be explained by relocation of the cells. These new data can be found in Figure 1F.

Our main tool to monitor osteoblast migration is the *bglap*:GFP transgenic line. Although *bglap* expression is downregulated during osteoblast dedifferentiation and thus also GFP levels eventually drop in the transgenic line, we can nevertheless use this line to trace osteoblasts, since GFP protein persists for up to three days in cells that shut down endogenous *bglap* and also *bglap*:GFP transgene transcription. While we have already shown this previously (Knopf et al., Dev Cell 2011, doi: 10.1016/j.devcel.2011.04.014), we have now also used RNAScope to confirm this. We analysed the expression of GFP on protein and RNA level in the *bglap*:GFP line. In *bglap*:GFP fish, in a mature segment in non-injured fins the regions close to the joints are devoid of cells expressing GFP (Figure 1G). Yet after amputation, we observe GFP+ cells in this distal part of segment -1 (Figure 1G, D). RNAscope in situ shows that these GFP+ cells are negative for *gfp* RNA (new data in Figure 1D). Thus, the observed fluorescence is due to the persistence of the GFP protein and not due to a potential upregulation of the transgene, which confirms that these GFP+ cells have migrated from the centre of the segment.

To confirm the specificity of the *bglap*:GFP line for mature osteoblasts, we performed immunofluorescence against Runx2 on 7 dpa regenerates, at a stage where blastema proliferation at the distal tip of the regenerate produces new osteoblast progenitors, while in more proximal (older) regions osteoblasts have already started to differentiate and new bone matrix has formed. We found that Runx2 is expressed in distal regions in pre-osteoblasts, while *bglap*:GFP is only expressed in proximal regions in osteoblasts which do not express Runx2. Thus, *bglap*:GFP is activated in mature osteoblasts and the population does not include osteoblast precursor cells. These new data are found in Figure 2 —figure supplement 2B. In addition, we now also show by -RNAScope that *bglap*:GFP+ cells in segment -2 (the segment not affected by dedifferentiation) do not express the pre-osteoblast marker *runx2a* nor *cyp26b1*, which is upregulated in dedifferentiated osteoblasts. These additional data confirm that *bglap*:GFP specifically labels differentiated osteoblasts and does not include osteoblast precursors.

We have restructured the first paragraphs of the results to better describe our choice of markers and tools and describe the new data in the revised manuscript on page 6 as follows:

“(…) Using RNAScope in situ hybridization, we can now show that downregulation of bglap occurs in a graded manner and that entpd5 expression is similarly downregulated during dedifferentiation (Figure 1B, C). At 1 day post amputation (1 dpa), expression of entpd5 and bglap remains high in segment -2, but gradually decreases towards the amputation plane and is almost entirely absent from segment 0, with entpd5 downregulation being more pronounced (Figure 1B, C). While RNA expression of these genes is downregulated within hours after injury, GFP or Kaede fluorescent proteins (FPs) expressed in bglap or entpd5 reporter transgenic lines persist for up to three days, even though transgene transcription is shut down rapidly as well (Knopf et al., 2011). We can confirm these earlier findings using the more sensitive RNAScope in situs. In bglap:GFP transgenics at 2 dpa, gfp RNA and GFP protein colocalized to the same cells in segment -2, where osteoblasts do not dedifferentiate (Figure 1D). In contrast, in the distal segment -1 GFP protein was present, but barely any gfp transcript could be detected (Figure 1D). Thus, persistence of FPs in reporter lines can be used for short-term tracing of dedifferentiated osteoblasts (Figure 1E). At 1 dpa, bglap:GFP+ cells upregulated expression of the pre-osteoblast marker runx2a and of cyp26b1, an enzyme involved in retinoic acid signalling (Blum and Begemann, 2015), which regulates dedifferentiation (Figure 1 —figure supplement 1A, B). Both markers were exclusively upregulated in segment -1 and segment 0 at 1 dpa, but were absent in segment -2. Together, these data show that osteoblasts in segment -1 and segment 0 lose expression of mature markers and gain expression of dedifferentiation markers.”

3) Additional results are needed to support the claim that the complement system is required for injury–induced directed osteoblast migration including expression analyses of the central complement components C3 and C5 at the amputation site and showing whether c5ar1 and c3ar are solely expressed in osteoblasts, or rather broadly within tissue lining the hemirays.

Unfortunately, we lack the tools to detect the C3 and C5 precursor proteins or the mature cleavage products of the complement factors, which mediate the biological function of the cascade (e.g. antibodies against the zebrafish proteins / peptides). However, we analysed expression of the RNAs coding for the precursors of the complement factors *c5* and the six zebrafish paralogs of *c3* using qRT-PCR on liver, non-injured fins and fins at 6 hpa (samples derived from segment -1 plus segment 0). These new data can be found in Figure 5B. Compared to the expression levels in the liver, expression in non-injured fins could hardly be detected. Interestingly, *c5* and *c3a.5* levels were upregulated in injured fins, but compared to the expression in the liver still only slightly, e.g. *c5* is about 17 Ct values (2 to the power of 17 = 130000 times) more highly expressed in the liver than in the injured fin. These results are consistent with the idea that the majority of complement factors that are activated after injury is derived from precursors that are expressed in the liver and are distributed via the circulation to the fin, as is considered standard for the complement system. Interestingly, however, local injury-induced expression in the fin might contribute precursor proteins as well.

While we had already shown that *c5aR1* is expressed in osteoblasts*,* we have now added additional RNAscope in situ analysis for *c5aR1* showing that the receptor is also expressed in other cell types in the fin (new data in Figure 5 —figure supplement 1A). We have also attempted RNAScope for *c3aR1, c5a* and *c3a.1* in fins, but these turned out to not produce any specific stainings, thus the results of these experiments remained inconclusive and we have not included them in the manuscript. However, we have established fluorescent activated cell sorting from *bglap*:GFP transgenic fins, which gives us an additional tool to analyse to which extent expression is specific to osteoblasts. By qRT-PCR analysis we found that *c5aR1* and *c3aR* are expressed in both GFP+ osteoblasts and other cells that are GFP– (these will mainly represent epidermis and fibroblasts, to a lesser extent endothelial and other cell types). These new data can be found in Figure 5 —figure supplement 1B.

While our qRT-PCR data and the *c5aR1* RNAScope results show that the complement receptors are not specifically expressed in osteoblasts, we do not consider this result to be in conflict with our model that the complement system regulates osteoblast migration. Other cell types migrate after fin amputation as well, which is best described for epidermal cells (Chen et al., Dev Cell 2016, 10.1016/j.devcel.2016.02.017), but likely also occurs for fibroblasts (Poleo et al., DevDyn 2001, doi: 10.1002/dvdy.1152), and it is conceivable that the complement system plays a role in regulating these events as well.

Overall, our new data support our conclusion that the complement system is an important regulator of osteoblast migration in vivo, since the receptors are present in osteoblasts, while systemic and local expression can provide the precursors for injury-induced production of the activated factors that might act as guidance cues.

4) It is unclear whether each step of cell elongation and alignment, cell migration, cell dedifferentiation and regenerative response, is required for fin regeneration following amputation. In the absence of direct evidence for the requirement of migration or dedifferentiation for the overall success of fin regeneration, the limitations should be more clearly stated, and the title modified.

We have modified the title and abstract to avoid overstating the requirement of the particular responses to successful regeneration. Furthermore, we have stated the limitations of our study more clearly in the discussion.

The title now reads:

“Zebrafish fin regeneration involves generic and regeneration-specific osteoblast injury responses”

In the discussion we state the limitations on page 21 as follows:

“Unfortunately, currently existing tools to block dedifferentiation are either mosaic (activation of NF- κB signalling using the Cre-lox system) or cannot be targeted to osteoblasts alone (treatment with retinoic acid). Due to these limitations in our assays, we can currently not test what consequences specific, unmitigated perturbation of osteoblast dedifferentiation has for overall fin / bone regeneration. Conversely, the interventions presented here that specifically perturb osteoblast migration are limited as they act only transiently, that is they can severely delay, but not fully block migration. Furthermore, while interference with actomyosin dynamics reduces regenerative growth, we cannot distinguish whether this is caused by the inhibition of osteoblast migration or due to other more direct effects on cell proliferation and tissue growth. Thus, an unequivocal test of the importance of osteoblast migration for bone regeneration requires different tools.”

5) The word trauma should be clarified as the authors do not provide evidence for a non–healing phenotype in their trauma model.

We apologize if our use of the term trauma has caused confusion. We have simply used it interchangeably with “injury”. We have now removed all references to “trauma” in the text.

6) The conclusions referring to the relevance of the work to bone regeneration should be toned down and more focused on fin regeneration.

We have rephrased the conclusion to have it more centred on bone regeneration in the fin.

The relevant parts of the discussion on page 25 now read as follows:

“In conclusion, our findings support a model in which zebrafish fin bone regeneration involves both generic and regeneration-specific injury responses of osteoblasts. Morphology changes and directed migration towards the injury site as well as dedifferentiation represent generic responses that occur at all injuries even if they are not followed by regenerative bone formation. While migration and dedifferentiation can be uncoupled and are (at least partially) independently regulated, they appear to be triggered by signals that emanate from all bone injuries. In contrast, migration off the bone matrix into the bone defect, formation of a population of (pre-) osteoblasts and regenerative bone formation represent regeneration-specific responses that require additional signals that are only present at distal-facing injuries. The identification of molecular determinants of the generic vs regenerative responses will be an interesting avenue for future research.”

7) The manuscript is overall well written but could be improved to be less technical to help the readers, and the discussion should be reduced in length.

We have edited the manuscript, in particular we have tried to explain the background and logic of our setup and tools at the beginning of the results. We have also shortened the discussion.

Reviewer #1 (Recommendations for the authors):To determine whether each step of cell elongation and alignment, cell migration, cell dedifferentiation and regenerative response, is required for fin regeneration following amputation the authors discuss p14 that other tools will be needed. However, later time points could be assessed in this study, not only to show the long–term effects of the treatment on healing but also to show how healing eventually resolves in the two models used here. Did the authors assess the long–term effects of inhibitor treatments on the regeneration process? Does blocking the complement system or actomyosin prevent regeneration? Are both generic (migration, dedifferentiation) and regenerative growth required for regeneration? Since there are limitations in the approaches used, the authors should modify the title and replace the word "requires". The use of the term "require" should be reconsidered also in the abstract and throughout the manuscript if the requirement for healing or a specific process is not demonstrated.

As a read-out for long-term effects of blocking the complement system or actomyosin, we have analysed two additional aspects. First, we analysed if the upregulation of Runx2 in segment 0 at 2 dpa is affected. We consider this another, but later readout for osteoblast dedifferentiation in addition to the *bglap* downregulation at 1 dpa, which we also show is not affected by actomyosin or complement inhibition (Figure 3F and Figure 5F). Neither actin nor complement interference had an effect on Runx2 upregulation. These new data can be found in Figure 3 —figure supplement 1B and Figure 5 —figure supplement 3B. These data confirm that these interventions do not block osteoblast dedifferentiation.

Second, we measured the length of the regenerate at 3 dpa as read-out for regenerative growth. Complement inhibition had no effect on regenerate length (new Figure 5 —figure supplement 3C). However, while osteoblast migration is severely impaired early after fin amputation upon complement inhibition, osteoblasts eventually do migrate to the amputation plane. We attempted to use higher doses of the drugs and the C3aR and C5aR1 inhibitors in combination, but these interventions turned out to be toxic and were thus inconclusive. Thus, the absence of regenerate growth defects upon complement inhibition cannot be interpreted to indicate that osteoblast migration is not required for fin regeneration.

In contrast, actomyosin inhibition did impair regenerative growth, (new data in Figure 3 —figure supplement 1C). Unfortunately, since actomyosin has essential roles in many cellular processes we cannot distinguish whether the impaired growth upon actomyosin inhibition is an indirect consequence of failed osteoblast migration or due to a direct role of actomyosin dynamics in cell proliferation and growth. Thus, the situation remains as we have explained in the discussion, that thorough tests of the individual requirements of the different osteoblast injury responses for overall regeneration will require additional tools. While this represents a limitation or our study, we still consider our main finding, that different injury responses can be uncoupled and are independently regulated, an important advance in our understanding of regeneration. We have revised the manuscript and replaced the word “require” with “involve”, e.g. in the title and abstract, which now read:

“Zebrafish fin regeneration involves generic and regeneration-specific osteoblast injury responses”

“Successful regeneration requires the coordinated execution of multiple cellular responses to injury. In amputated zebrafish fins, mature osteoblasts dedifferentiate, migrate towards the injury and form proliferative osteogenic blastema cells. We show that osteoblast migration is preceded by cell elongation and alignment along the proximodistal axis, which require actomyosin, but not microtubule turnover. Surprisingly, osteoblast dedifferentiation and migration can be uncoupled. Using pharmacological and genetic interventions, we found that NF-ĸB and retinoic acid signalling regulate dedifferentiation without affecting migration, while the complement system and actomyosin dynamics affect migration but not dedifferentiation. Furthermore, by removing bone at two locations within a fin ray, we established an injury model containing two injury sites. We found that osteoblasts dedifferentiate at and migrate towards both sites, while accumulation of osteogenic progenitor cells and regenerative bone formation only occur at the distal-facing injury. Together, these data indicate that osteoblast dedifferentiation and migration represent generic injury responses that are differentially regulated and can occur independently of each other and of regenerative growth. We conclude that successful bone regeneration appears to involve the coordinated execution of generic and regeneration-specific responses of osteoblasts to injury.”

Can the authors provide evidence of the source of C3 and C5, the central components of the complement system during fin regeneration? Figure 4 shows RNAscope in situ detection of c5aR1. Analysis of c3 and c5 using the same technique should be included.

Unfortunately, we lack the tools to detect the C3 and C5 precursor proteins or the mature cleavage products of the complement factors, which mediate the biological function of the cascade (e.g. antibodies against the zebrafish proteins / peptides). However, we analysed expression of the RNA coding for the precursors of the complement factors *c5* and the six zebrafish paralogs of *c3* using qRT-PCR on liver, non-injured fins and fins at 6 hpa (samples derived from segment -1 plus segment 0). These new data can be found in Figure 5B. Compared to the expression levels in the liver, expression in non-injured fins could hardly be detected. Interestingly, *c5* and *c3a.5* levels were upregulated in injured fins, but compared to the expression in the liver still only slightly, e.g. *c5* is about 17 Ct values (2 to the power of 17 = 130000 times) more highly expressed in the liver than in the injured fin. These results are consistent with the idea that the majority of complement factors that are activated after injury is derived from precursors that are expressed in the liver and are distributed via the circulation to the fin, as is considered standard for the complement system. Interestingly, however, local, injury-induced expression might contribute precursor proteins as well.

Regarding the trauma model, it is unclear whether the gaps on each side of the remaining segment eventually heal. Does the gap on the proximal side of the remaining segment heal via regeneration occurring on the distal end of the next segment? The term "trauma" should be replaced by another term. Further, the term "trauma" appears to be used in the first paragraph of the discussion when referring to amputation and in other parts of the discussion when referring to bone injury.

In the hemiray injury model, the proximal side does indeed heal via regeneration form the distal end of the proximally located segment. We have added this information to the manuscript on page 16 as follows:

“Similarly, new bone matrix was formed at the distal side of the remaining proximal segment (arrowhead in Figure 6A and data not shown), but not at the proximal injury site.”

We apologize if our use of the term trauma has caused confusion. We have simply used it interchangeably with “injury”. We have now removed all references to “trauma” in the text.

The Discussion section is very long and there are many redundancies with the Results section. The main findings and conclusions could be stated more briefly, and the discussion points presented in a more concise way. As indicated in the public review, the last paragraph of the discussion should be focused on "fin regeneration" and avoid concluding on "bone regeneration" and "bone formation".

We have shortened the discussion to avoid redundancies, and have rephrased the conclusion to have it more centred on bone regeneration in the fin.

Reviewer #2 (Recommendations for the authors):– Page 4, line 6: I am not sure if "exoskeletal" is by definition the right term in this context.

We thank the reviewer for the correction. We have rephrased the text on page 4 as follows:

“The fin skeleton consists of dermal (directly ossifying) skeletal elements, the fin rays or lepidotrichia, which make up the largest part of the fin, and endochondral parts close to the body.”

– What is the p–value in Figure 2A (left) between the width/length ratio of cytochalasin–treated osteoblasts from segments –1 and –2? It looks as if the cells of segment –1 are somewhat elongated even though they have been exposed to the drug. Could the authors please comment on this?

Upon cytochalasin treatment, osteoblasts in segment -1 are indeed elongated compared to segment -2 (p = 0.015, note that the data is now in Figure 3A). For the drug treatments affecting actomyosin, we had to carefully titrate drug concentrations such that the overall well-being of the fish was not affected. Thus, our hypothesis is that with the drug concentration we used, actin dynamics are impaired but not fully abolished. Therefore, one can still observe a cell shape change, but it is significantly decreased compared to non-treated fish.

– The red labeling of data points for the drug–treated conditions in Figure 2A, B and C might be difficult to be detected by color–blind readers. This should be adjusted accordingly.

As suggested by the reviewer, we have changed the color code.

– An image showing cyp26b1 RNAscope in situ hybridization on which the graphs of Figure 5H are based should be shown somewhere – maybe in the dedicated Excel sheet of raw data.

We have added a representative image of *cyp26b1* RNAscope in situ hybridization (new data in Figure 6 —figure supplement 1B).

– Suggestion for a further experiment (not required for acceptance from my side!!!): I think it would be nice to learn whether C3a and C5a are sufficient to drive directed osteoblast migration along bony elements. To test that one could try to implant a bead loaded with recombinant C3a or C5a somewhere along a fin ray to see whether this ectopic signaling source attracts osteoblasts. Would this be doable? Recombinant complement proteins appear to be commercially available.

This is a great idea, and we have actually tried this by providing an ectopic source of the ligands next to the bony rays. However, it is technically very challenging to implant beads into the rather thin tissue between rays. Thus, we have not yet been able to derive conclusions from this experiment.

Reviewer #3 (Recommendations for the authors):1. Title:– I feel that the current title does not capture the essence of the authors' major findings.

In our study, we analyse several aspects (dedifferentiation, migration, proliferation) of osteoblast responses to amputation. We feel that the independence of these responses and the fact that dedifferentiation and migration can occur independently of regenerative bone growth are the main novel findings, which we tried to highlight in the title. However, since we agree with comments by several reviewers that we should be more cautious in proclaiming that all injury responses are “required”, we have replaced this term in the title with “involves”, the title now reads:

“Zebrafish fin regeneration involves generic and regeneration-specific osteoblast injury responses”

2. Abstract:– The abstract is oversimplified and overstated, omitting important details. The authors should clearly state that they used a bglap:GFP strain to support their conclusions on osteoblast differentiation and migration.

As described above, we have several lines of additional evidence beyond the *bglap*:GFP line supporting our conclusions about osteoblast migration and its independence from dedifferentiation. We thus feel that we can use somewhat more general language in the abstract.

To avoid overstating the requirement of the particular responses to successful regeneration, we have modified the abstract. The last sentence now reads as follows:

“We conclude that successful bone regeneration appears to involve the coordinated execution of generic and regeneration-specific responses of osteoblasts to injury.”

3. Introduction:– The authors make significant statements on osteoblast dedifferentiation. This is based on a naïve assumption that bglap:GFP is solely active in mature osteoblasts. However, it is clear from the authors' data that a substantial fraction of GFP+ cells are in proliferation (Figure 4 S1B). It seems that bglap:GFP is also expressed by osteoblast precursors. Specificity of bglap:GFP to mature osteoblasts is a major concern.

To address these comments, we have performed several additional experiments as described below. In addition, we would like to refer the reviewer to our previous papers, where we have analysed the process of osteoblast dedifferentiation (Knopf et al., Dev Cell 2011, doi: 10.1016/j.devcel.2011.04.014; Geurtzen et al., Development 2014, doi: 10.1242/dev.105817; Mishra et al. Dev Cell 2020, doi: 10.1016/j.devcel.2019.11.016). Using transgenic reporters and immunofluorescence we have shown in these previous papers that osteoblasts in the non-injured fin express Bglap but not the pre-osteoblast marker Runx2 (and are thus by our definition differentiated). We apologize if we failed to explain the logic of our approach in this manuscript, we have restructured the results to clarify these, as indicated below.

We have also performed the following additional experiments.

1) To confirm the specificity of the *bglap*:GFP line for mature osteoblasts, we have performed three experiments:

a) Immunofluorescence against Runx2 on 7 dpa regenerates, at a stage where blastema proliferation at the distal tip of the regenerate produces new osteoblast progenitors, while in more proximal (older) regions osteoblasts have already started to differentiate and new bone matrix has formed. We found that Runx2 is expressed in distal regions in pre-osteoblasts, while *bglap*:GFP is only expressed in proximal regions in osteoblasts which do not express Runx2. Thus, formation of new bony segment during regenerative growth, *bglap*:GFP is activated in mature osteoblasts and the population does not include osteoblast precursor cells. These new data are found in Figure 2 —figure supplement 2B.

b) We have refined and expanded our methods and are now able to determine the expression patterns of markers of the osteoblast differentiation status with single cell resolution using RNAScope in situ hybridization. Using this, we can now show that at 1 day post amputation, in segment -2 of the fin stump, which represents a segment equivalent to the non-injured state, since no dedifferentiation occurs here, *bglap*:GFP+ cells do not express endogenous *runx2a*. These new data are found in Figure 1 —figure supplement 1A.

c) Using RNAScope, we can show that *cyp26b1*, a gene associated with dedifferentiated osteoblasts, is likewise not detected in *bglap*:GFP+ cells in segment -2 at 1 dpa (new data in Figure 1 —figure supplement 1B).

Together, these data confirm that the bglap:GFP line is specific for differentiated osteoblasts, and does not label osteoblast progenitors.

See the response to issue 2 below for how we describe these new data in the revised version of the manuscript.

2) Regarding the proliferation of *bglap*:GFP osteoblasts: In the experiment the reviewer refers to (now Figure 5 —figure supplement 3A), we make use of the persistence of the GFP protein in the *bglap*:GFP line to detect dedifferentiated osteoblasts. Thus, at the time of analysis, when these GFP+ cells proliferate, they are not differentiated anymore. We can show this as follows:

Although *bglap* expression is downregulated during osteoblast dedifferentiation and thus also GFP levels eventually drop in the transgenic line, we can nevertheless use this line to trace osteoblasts, since GFP protein persists for up to three days in cells that shut down endogenous *bglap* and also *bglap*:GFP transgene transcription. While we have already shown this previously (Knopf et al., Dev Cell 2011, doi: 10.1016/j.devcel.2011.04.014; Geurtzen et al., Development 2014, doi: 10.1242/dev.105817; Mishra et al. Dev Cell 2020, doi: 10.1016/j.devcel.2019.11.016), we have now also used RNAScope to confirm this. We analysed the expression of GFP on protein and RNA level in the *bglap*:GFP line. In *bglap*:GFP fish, in a mature segment in non-injured fins the regions close to the joints are devoid of cells expressing GFP (Figure 1G). Yet after amputation, we observe GFP+ cells in this distal part of segment -1 (Figure 1G, D). RNAscope in situ shows that these GFP+ cells are negative for *gfp* RNA (new data in Figure 1D). Thus, the observed fluorescence is due to the persistence of the GFP protein and not due to a potential upregulation of the transgene (Figure 1E).

Importantly, we have now also added data describing the proliferative state of *bglap*:GFP+ osteoblasts. First, in the non-injured fin, *bglap*:GFP+ cells are non-proliferative (new data in Figure 5 —figure supplement 2B). After amputation, proliferation can be detected in GFP+ cells at 2 dpa (Figure 5 —figure supplement 2B), and proliferation is restricted to segment -1 and segment 0 (new data in Figure 5 —figure supplement 2C). As we show in Figure 1B, at 2 dpa, dedifferentiation as defined by *bglap* downregulation is not complete in segment -1, rather here a mixture of cells with different *bglap* levels are found. We have thus combined EdU labelling with RNAscope against *bglap* in segment -1 to analyse to which extent *bglap* and EdU anticorrelate. These data show that EdU is hardly ever incorporated into cells expressing high levels of *bglap*, while the majority of the proliferating osteoblasts are dedifferentiated, as they express only low levels of *bglap* (new data in Figure 5 —figure supplement 2D). Together, these data show that mature osteoblasts are non-proliferative, and upon amputation, when they are dedifferentiated, they become proliferative. Thus, the absence of proliferation in *bgla*p:GFP+ cells in the non-injured fin adds to the evidence that this line is specific for mature osteoblasts, but due to the persistence of the GFP protein it can be used to analyse dedifferentiated osteoblasts.

These data are described on page 14 of the manuscript as follows:

“In the non-injured fin, bglap:GFP+ osteoblasts are non-proliferative, but upon amputation osteoblasts proliferate at 2 dpa (Figure 5 —figure supplement 2A, B). Proliferation is restricted to segment -1 and segment 0 (Figure 5 —figure supplement 2C), and RNAscope in situ analysis of bglap expression revealed that the majority of EdU+ osteoblasts have strongly downregulated bglap (Figure 5 —figure supplement 2D). Inhibition of C5aR1 with PMX205 had no effect on osteoblast proliferation in segment -1 at 2 dpa (Figure 5 —figure supplement 3A). Furthermore, upregulation of Runx2 was not changed by PMX205 treatment (Figure 5 —figure supplement 3B), and regenerative growth was not affected in fish treated with either W54011, PMX205 or SB290157 (Figure 5 —figure supplement Figure 3C). We conclude that the complement system specifically regulates injury-induced osteoblast migration, but not osteoblast dedifferentiation or proliferation in zebrafish.”

3) To support our conclusion that osteoblasts migrate, we performed time-lapse imaging using a transgenic line expressing the photoconvertible protein kaede in osteoblasts (*entpd5*:kaede). Local photoconversion of only the proximal half of a segment allowed us to trace these photoconverted osteoblasts. This revealed that converted cells appear in the distal part of the segment within 1 dpa, which can only be explained by relocation of the cells. These new data can be found in Figure 1F and they are described on page 7 of the revised manuscript as follows:

“To trace osteoblasts, we used the transgenic line entpd5:kaede (Geurtzen et al., 2014), in which Kaede fluorescence can be converted from green to red by UV light (Ando et al., 2002). We photoconverted osteoblasts in the proximal half of segment -1, while osteoblasts in the distal half remained green (Figure 1F). At 1 dpa, red osteoblasts were found in the distal half (Figure 1F), showing that photoconverted osteoblasts had relocated distally.”

– Page 5, Line 5: Osteoblasts do not migrate to the resorbed sites. Osteoblast precursors (or pre–osteoblasts) do. Thiel 2018 review article does not appear to be particularly supportive.

We thank the reviewer for the correction. We have corrected it and added another reference:

“For bone formation, osteoblast precursors migrate to the resorbed sites (Dirckx, Van Hul and Maes, 2013)”

4. Results:– Page 6, Line 14: Perhaps the authors should explain in more detail about the kaedeGreen to Red system, and why they needed this approach.

We have expanded the description of the photoconvertible kaede system. It is first introduced now on page 7 (describing a new experiment), where it reads as follows:

“To trace osteoblasts, we used the transgenic line *entpd5*:kaede (Geurtzen et al., 2014), in which Kaede fluorescence can be converted from green to red by UV light (Ando et al., 2002).”

In the experiment the reviewer is referring to, we used a double transgenic line expressing both GFP and kaede in osteoblasts (Video 1, Figure 2 —figure supplement 1A). We deliberately only partly converted kaedeGreen to kaedeRed, which resulted in different hues for each osteoblast. This distinct colouring facilitates identification of individual cells during repeated live imaging using a confocal microscope. Due to the curved surface of a hemiray and to counteract potential movements along the z-axis of the fin during image acquisition, several z-slides had to be recorded at each timepoint. As a consequence, there is a considerable time of recording, increasing photobleaching over time. Having two fluorescent proteins being expressed in the cells gave a higher intensity signal in the beginning, allowing lower laser power and exposure time, and thus reduced photobleaching. The description now reads as follows (page 8):

“To observe osteoblast behaviour at single cell resolution in live fish after fin amputation, we generated double transgenic fish expressing bglap:GFP (Knopf et al., 2011)and entpd5:kaede in mature osteoblasts. Expression of two fluorescent proteins increased signal for repeated imaging, and partial photoconversion of kaedeGreen into kaedeRed resulted in a different colouring for each osteoblast, which facilitated tracking of morphology changes at single cell resolution.”

– Page 7, Line 13–16: I do not understand why the authors interpret these data as directed active migration behavior towards the injury. What they showed here is merely cell shape changes.

The formation of protrusions orientated in the direction of (nascent) movement is an essential step during cell migration. Our data show that mature osteoblasts in the non-injured fin are roundish without such directed protrusion. In contrast, after amputation we see an elongation of osteoblasts in segment -1 in the direction of migration. We agree that such protrusions on their own would not be sufficient evidence of active migration. Yet, in combination with our other evidence (relocation of cells shown by *bglap*:GFP and *entpd5*:Kaede tracing via repeated imaging, live videos showing movement of osteoblasts relative to their surroundings, see the response to “Weaknesses”, issue 3 above), we interpret this morphology changes as being consistent with acquisition of active migratory behaviour. To better elucidate the relation between cell shape changes and migration, we have re-arranged the order in which we present our data. We now show the relocation / migration of osteoblasts first (Figure 1D – I), and then the cell shape changes (Figure 2A-C, Figure 2 —figure supplement 1). This highlights that only in the segments where osteoblasts migrate, they elongate. Furthermore, we have rephrased the sentence. It now reads as follows (page 9):

“We interpret the elongation and orientation along the proximodistal axis and the formation of long-lived protrusions along this axis as events that prime osteoblasts for directed active migration towards the injury.”

– Page 7, Line 19: This data could be alternatively interpreted as the upregulation of bglap:GFP by invading cells (not previously bglap+), rather than the invasion of bglap:GFP+ cells. It would be essential to determine the half–life of GFP to assume cell invasion. The authors would need a more long–lasting histone–bound GFP reporter strain to conclusively define cell invasion.

Please see our response above. There we describe new experiments where we detect *gfp* RNA and GFP protein to show that appearance of GFP+ cells is indeed due to invasion, not upregulation of transgene transcription. See new data in Figure 1D, E.

– Page 8, Line 3–16: This paragraph is very difficult to read because of long sentences. They should simplify and rephrase the sentences.

We have rephrased this paragraph and shortened the sentences. It now reads as follows (page 9):

“Fins grow by the addition of new segments distally, and in the distal-most, youngest segment, bglap:GFP is not expressed (Knopf et al., 2011). Similarly, in the regenerating fins increasing numbers of GFP+ cells can be detected in more proximally located, older segments, reflecting the progressive differentiation of osteoblasts with time (Figure 2 —figure supplement 2A). In less mature segments, the pre-osteoblast marker Runx2 can be detected, but its expression does not overlap with bglap:GFP expression in mature osteoblasts (Figure 2 —figure supplement 2B). Importantly, in newly regenerated segments that start to upregulate bglap expression, all bglap:GFP+ cells appear in the centre of the segments; we could not observe GFP+ cells within joints (Figure 2 —figure supplement 2A). This suggests that bglap:GFP+ osteoblasts from older segments do not migrate into less mature segments during formation of new segments in the course of fin growth. Rather, the mature osteoblast population in a segment is formed via differentiation of osteoblasts at the position within the segment where they were formed during segment formation. In contrast, within 2 days after fin amputation, bglap:GFP+ cells appeared in the fin stump within the joint between segment -1 and segment 0 (Figure 2D, yellow arrowhead), indicating that GFP+ osteoblasts from segment -1 crossed the joint during their migration towards the amputation plane. Thus, migration of mature osteoblasts observed after fin amputation or bone fracture (Geurtzen et al., 2014) appears to represent a specific early response to injury, while addition of new bony segments at the distal tip of growing fins during ontogeny or regeneration does not involve migration of differentiated osteoblasts.”

– Page 9, Line 1: The authors should move the bglap RNAscope in situ data to the main figure, as this is an important piece of information.

We agree that these data were somewhat hidden in the Supplement, we have now moved them to the re-designed Figure 1B.

– Page 9, Line 6: I don't understand why reduced bglap mRNA can be interpreted as osteoblast dedifferentiation.

We define dedifferentiation as the reversion of a mature cell into an undifferentiated progenitor-like status. This involves the following characteristics: (1) the expression of markers of the differentiated state are downregulated; (2) early lineage markers are re-expressed; (3) the cells become proliferative; and (4) they have the ability to re-differentiate into mature cells. Based in this definition, the downregulation of an osteoblast-specific marker can be used as a read-out for osteoblast dedifferentiation. *Bglap* is an established marker for mature osteoblasts (Kaneto et al., 2016 doi.org/10.1186/s12881-016-0301-7¸ Yoshioka et al., 2021 doi: 10.1002/jbm4.10496; Kannan et al., 2020 doi: 10.1242/bio.053280; Sojan et al., 2022 doi.org/10.3389/fnut.2022.868805; Valenti et al., 2020 doi.org/10.3390/cells9081911). While we use downregulation of *bglap* expression as our main read-out for osteoblast dedifferentiation in our experimental interventions (actomyosin inhibition, retinoic acid treatment, complement inhibition), we have expanded our methods to characterize osteoblast dedifferentiation, and have re-arranged our manuscript to show these data in the beginning of the results.

Already in the previous version of the manuscript we have shown that endogenous *bglap* is strongly expressed in segment -2, (the segment that does not respond to fin amputation and thus represents the non-injured state), while it is downregulated in a graded manner in segment -1 and segment 0 (the segments where dedifferentiation happens). We have now moved this data to the re-designed Figure 1B. In addition to *bglap*, we can now show that *entpd5,* a gene required for bone mineralization, is strongly expressed in osteoblasts of segment -2, while it is massively downregulated in segment -1 and segment 0. These new data can be found in Figure 1C. Thus, *entpd5* is another differentiation marker whose loss characterizes osteoblast dedifferentiation. Importantly, we can confirm by RNAScope that the pre-osteoblast marker *runx2a* is absent in mature segments but is upregulated in segment 0 and segment -1 at 1 dpa (new data in Figure 1 —figure supplement 1A). Similarly, *cyp26b1*, an enzyme shown to regulate dedifferentiation, is upregulated in segment 0 and segment -1, but not expressed in segment -2. (new data in Figure 1 —figure supplement 1B). Furthermore, we have repeated all experiments where we have previously quantified dedifferentiation upon experimental interventions using downregulation of *bglap*:GFP (actomyosin inhibition, retinoic acid treatment, complement inhibition). We now can fully confirm the previous conclusions using the more rigorous quantification of dedifferentiation using RNAScope analysis of endogenous *bglap* levels. We have replaced all *bglap*:GFP data with the new *bglap* RNAScope data. These new data are found in Figure 3F, Figure 3 —figure supplement 1A, Figure 4B and Figure 5F.

Overall, we support our conclusion that osteoblasts dedifferentiate by the loss of the two differentiation markers *bglap* and *entpd5*, the upregulation of the pre-osteoblast marker *runx2a* and the dedifferentiation-associated gene *cyp26b1*, and the fact that osteoblasts become proliferative. We hope that the reviewer considers this sufficient evidence.

In mammals, the available literature relatively convincingly concludes that NF-κB signaling negatively regulates osteoblast differentiation (Yao et al., 2014, doi: 10.1002/jbmr.2108; Swarnkar et al., 2014 doi.org/10.1371/journal.pone.0091421, Chang et al., 2009, doi.org/10.1038/nm.1954). Yet in zebrafish osteoblasts, we have previously shown that NF-κB signaling is active in mature osteoblasts and needs to be downregulated for dedifferentiation to occur (Mishra et al., 2020, 10.1016/j.devcel.2019.11.016). Importantly, in our previous work we showed that at least during fin regeneration, NF-κB signalling is not involved in osteoblast differentiation (Mishra et al., 2020, 10.1016/j.devcel.2019.11.016). Specifically, osteoblasts in which Nf-kappaB signaling is enhanced or inhibited differentiate completely normally during the later stages of fin regeneration in the fin regenerate. Hence, our findings with the Nf-kappaB intervention studies done in this manuscript, where we look at osteoblasts in the stump within 1 dpa, cannot be explained by them affecting osteoblast differentiation.

For retinoic acid signalling, multiple roles in bone development and repair have been described in mammals. For zebrafish osteoblasts, it was shown that during the outgrowth phase of bone regeneration, retinoic acid negatively regulates osteoblast differentiation in the blastema (Blum & Begemann, 2015, 10.1242/dev.120204). Yet importantly, it also negatively controls the dedifferentiation of osteoblasts in the stump right after amputation (Blum & Begemann, 2015, 10.1242/dev.120204). Thus, the effect we observe at the early timepoints we analyse in our intervention studies (retinoic acid treatment) are due to the effect on osteoblast dedifferentiation.

We have added a short definition of dedifferentiation to the Results section (page 6). There it reads as follows:

“We have previously shown that osteoblasts dedifferentiate in response to fin amputation, that is they revert from a mature, non-proliferative state into an undifferentiated progenitor-like state, which includes loss of bglap expression and upregulation of the pre-osteoblast marker runx2 (Knopf et al., 2011; Geurtzen et al., 2014).”

In addition, we have restructured the results to describe our use of tools and the new data on page 6 of the revised manuscript as follows:

“Using RNAScope in situ hybridization, we can now show that downregulation of bglap occurs in a graded manner and that entpd5 expression is similarly downregulated during dedifferentiation (Figure 1B, C). At 1 day post amputation (1 dpa), expression of entpd5 and bglap remains high in segment -2, but gradually decreases towards the amputation plane and is almost entirely absent from segment 0, with entpd5 downregulation being more pronounced (Figure 1B, C). While RNA expression of these genes is downregulated within hours after injury, GFP or Kaede fluorescent proteins (FPs) expressed in bglap or entpd5 reporter transgenic lines persist for up to three days, even though transgene transcription is shut down rapidly as well (Knopf et al., 2011). We can confirm these earlier findings using the more sensitive RNAScope in situs. In bglap:GFP transgenics at 2 dpa, gfp RNA and GFP protein colocalized to the same cells in segment -2, where osteoblasts do not dedifferentiate (Figure 1D). In contrast, in the distal segment -1 GFP protein was present, but barely any gfp transcript could be detected (Figure 1D). Thus, persistence of FPs in reporter lines can be used for short-term tracing of dedifferentiated osteoblasts (Figure 1E). At 1 dpa, bglap:GFP+ cells upregulated expression of the pre-osteoblast marker runx2a and of cyp26b1, an enzyme involved in retinoic acid signalling (Blum and Begemann, 2015), which regulates dedifferentiation (Figure 1 —figure supplement 1A, B). Both markers were exclusively upregulated in segment -1 and segment 0 at 1 dpa, but were absent in segment -2. Together, these data show that osteoblasts in segment -1 and segment 0 lose expression of mature markers and gain expression of dedifferentiation markers.”

– Overall, the authors would need live–cell imaging and tracking to make these conclusions.

We have the following evidence for osteoblast migration:

*1) bglap*:GFP+ cells relocate from the centre of segments towards the amputation plane (after fin amputations) or towards both injuries in the hemiray model. In this revised manuscript we show that transgene expression is not upregulated in these regions, but that GFP fluorescence there must be due to relocation of cells in which GFP protein persists (new data in Figure 1D, E; see also response to “Weaknesses, issue 1” above)

2) Using the *entpd5*:kaede transgenic line, which is expressed in mature osteoblasts throughout segments, we have photoconverted only the proximal half of a segment, which allowed us to trace these photoconverted osteoblasts. This revealed that converted cells appear in the distal part of the segment within 1 dpa, which can only be explained by relocation of the cells. These new data can be found in Figure 1F.

3) Already in the previous version of the manuscript, we have performed live imaging to track single cell behaviour. Using double transgenic fish expressing both GFP and kaede in osteoblasts, we deliberately only partly converted kaedeGreen to kaedeRed, which resulted in different hues for each osteoblast. This distinct colouring facilitates observing single cells. Video 1 shows the directed movement of cell bodies relative to their surroundings within 2 hours (see also Figure 2 —figure supplement 1A).

4) Osteoblasts display the typical cell shape changes associated with active migration (elongation along the axis of migration, extension of dynamic protrusions), data in Figure 2.

Together, we think these are convincing data supporting the conclusion that osteoblasts actively migrate.

– Page 10, Line 7: If the authors think that a bglap:GFP strain can be used as live reporters for osteoblast dedifferentiation, that would significantly compromise the scientific rigor of the study, as it would not support any conclusions regarding cell migration and invasion.

Please see the response to “Weaknesses”, issue 1 above. As we have laid out there, *bglap*:GFP can be used for both short-term tracing of cells that used to be differentiated due to persistence of GFP protein after transgene shutdown during dedifferentiation, and as readout for dedifferentiation, since GFP levels eventually do drop as well. However, we agree that its use as a readout for dedifferentiation is complicated and not straightforward. Thus, we have repeated all experiments where we have previously quantified dedifferentiation upon experimental interventions using downregulation of *bglap*:GFP (actomyosin inhibition, retinoic acid treatment, complement inhibition). We now can fully confirm the previous conclusions using the more rigorous quantification of dedifferentiation using RNAScope analysis of endogenous *bglap* levels. These new data are found in Figure 3F, Figure 3 —figure supplement 1A, Figures 4B, Figure 5F.

– Page 10, Line 20: Osteoblast migration should not be mixed up with cell shape changes.

We agree that cell shape changes and relocation of cells relative to their surroundings are strictly not the same process. Yet our data support the idea that both are events associated with the same process, namely osteoblast migration. This recapitulating sentence refers to the data we presented regarding the effect of actomyosin and microtubuli drugs. For both interventions we have assayed osteoblast relocation (of *bglap*:GFP+ cells from the centre towards the distal end of the segment) and cell shape changes. We thus think that both assays together allow us to describe these data as affecting osteoblast “migration”. This sentence is now on page 12, line 1.

– Page 11, Line 12: The authors did not show data for osteoblast differentiation/dedifferentiation in their genetic intervention studies.

The genetic intervention was used to manipulate NF-κB signalling. There are established tools available to genetically interfere with NF-κB signalling, and we used such transgenic tools in our assays. NF-κB signalling can be inhibited by overexpression of the IkB super repressor (IkBSR) (Van Antwerp et al., 1996 doi: 10.1126/science.274.5288.787; Karra et al., 2015 DOI: 10.1073/pnas.1511209112.), while overexpression of a constitutively active IkB kinase (IKKca) results in overactivation of the pathway (Mishra et al., 2020 10.1016/j.devcel.2019.11.016). To specifically manipulate the pathway in osteoblasts, we employ the Cre-Lox system, using the established inducible line osx:CreER (Knopf et al., 2011, doi: 10.1016/j.devcel.2011.04.014). We have previously shown that such osteoblast-specific overexpression of IkBSR enhanced osteoblast dedifferentiation, while overexpression of IKKca impaired dedifferentiation (Mishra et al., 2020 10.1016/j.devcel.2019.11.016). Due to technical reasons, the assay to test the effect of the genetic interventions on dedifferentiation cannot be combined with the read-out for osteoblast cell shape. Yet, since we have shown the effect of these tools on dedifferentiation in our previous paper, we did not repeat these assays for this manuscript.

– Page 11, Line 19: An alternative explanation is that retinoic acid and NF–kB signaling simply promote osteoblast differentiation, without anything to do with dedifferentiation.

In mammals, the available literature relatively convincingly concludes that NF-κB signaling negatively regulates osteoblast differentiation (Yao et al., 2014, doi: 10.1002/jbmr.2108; Swarnkar et al., 2014 doi.org/10.1371/journal.pone.0091421, Chang et al., 2009, doi.org/10.1038/nm.1954). Yet in zebrafish osteoblasts, we have previously shown that NF-κB signaling is active in mature osteoblasts and needs to be downregulated for dedifferentiation to occur (Mishra et al., 2020, 10.1016/j.devcel.2019.11.016). Importantly, in our previous work we showed that at least during fin regeneration, NF-κB signalling is not involved in osteoblast differentiation (Mishra et al., 2020, 10.1016/j.devcel.2019.11.016). Specifically, osteoblasts in which Nf-kappaB signaling is enhanced or inhibited differentiate completely normally during the later stages of fin regeneration in the fin regenerate. Hence, our findings with the Nf-kappaB intervention studies done in this manuscript, where we look at osteoblasts in the stump within 1 dpa cannot be explained by them affecting osteoblast differentiation.

For retinoic acid signalling, multiple roles in bone development and repair have been described in mammals. For zebrafish osteoblasts, it was shown that during the outgrowth phase of bone regeneration, retinoic acid negatively regulates osteoblast differentiation in the blastema (Blum & Begemann, 2015, 10.1242/dev.120204). Yet importantly, it also negatively controls the dedifferentiation of osteoblasts in the stump right after amputation (Blum & Begemann, 2015, 10.1242/dev.120204). Thus, the effect we observe at the early timepoints we analyse in our intervention studies (retinoic acid treatment) are due to the effect on osteoblast dedifferentiation

– Page 12, Line 1–2: These conclusions are stretched and not supported by experimental data. The argument for the decoupling of dedifferentiation and migration is not well developed.

We politely disagree with the reviewer as we do think that our data support the hypothesis that dedifferentiation and migration are not coupled in the sense that they are necessary consequences that are triggered by a single event, namely injury, and thus always occur together. We base our hypothesis on the following observations: we define molecular interventions in which (1) osteoblast elongation and migration are impaired, yet dedifferentiation is not affected (interference with complement and actomyosin systems, Figure 3), and (2) interventions where dedifferentiation is inhibited, yet osteoblasts still elongate and migrate (Nf-kappaB and retinoic acid signaling, Figure 4). If these two aspects were interdependent, one would expect the mitigation of one inevitably affecting the strength of the other. Yet we do not see such an effect but instead one can be diminished without the other being changed. Therefore, we conclude that they are independently regulated. However, we have rephrased the sentence to attenuate our conclusions about the independency of dedifferentiation and migration. It now reads as follows:

“Together, these data suggest that osteoblast dedifferentiation and migration are independently regulated, as one response can be impaired without affecting the other. Furthermore, they indicate that osteoblast dedifferentiation is not a prerequisite for migration.”

– Page 13, Line 8: As many as 20% of bglap:GFP+ cells are EdU+ therefore in proliferation. How would the authors explain this?

Please see the response to “Weaknesses”, issue 1, our point 2 above. In brief, the GFP+ cells analysed here are already dedifferentiated, yet GFP protein persists.

5. Discussion:– The authors should clearly indicate their definition of osteoblasts in this study (i.e. bglap:GFP+ cells).

We state our definition of differentiated osteoblasts in the introduction as follows (page 4):

“…osteoblasts close to the amputation plane dedifferentiate, that is they downregulate the expression of the mature osteoblast marker bglap, upregulate pre-osteoblast markers like runx2 and become proliferative…*”*

And in the results as follows (page 6):

“We have previously shown that osteoblasts dedifferentiate in response to fin amputation, that is they revert from a mature, non-proliferative state into an undifferentiated progenitor-like state, which includes loss of bglap expression and upregulation of the pre-osteoblast marker runx2*.”*

And in the discussion again as follows (page 20):

“In addition, osteoblasts dedifferentiate, that is they downregulate expression of differentiation markers like *bglap* and initiate expression of the pre-osteoblast marker *runx2*.”

– Osteoblast dedifferentiation should be rephrased as reduced bglap expression.

We agree that the term “dedifferentiation” is unfortunately loosely defined in the literature, in particular in relation to the question whether a process described as “dedifferentiation” would include a fate reversal of cells. However, since most instances of naturally occurring dedifferentiation during unperturbed regenerative events involve reversal of differentiated cells to a proliferative, progenitor-like state in which the cells remain lineage restricted, we favour a definition as follows:

Dedifferentiation is the reversion of a mature cell into an undifferentiated progenitor-like status. This involves the following characteristics: (1) the expression of differentiation markers is downregulated; (2) early lineage markers are re-expressed; (3) the cells become proliferative; and (4) they have the ability to re-differentiate into mature cells.

Already in the previous version of the manuscript, in the first mention of dedifferentiation in the discussion, we define dedifferentiation as loss of bglap and gain of the pre-osteoblast marker runx2. It reads as follows (page 20):

“In addition, osteoblasts dedifferentiate, that is they downregulate expression of differentiation markers like *bglap* and initiate expression of the pre-osteoblast marker *runx2*.”

– Page 18, Line 19: Osteoblast differentiation, instead of osteoblast dedifferentiation.

We have to respectfully disagree with the reviewer, as we do indeed mean dedifferentiation. As explained in detail in our answer to the public review, point 2, in the zebrafish fin, after amputation both pathways (NF-κB and retinoic acid) regulate osteoblast dedifferentiation, which was shown previously by us and other labs. This sentence is now page 20:

“Retinoic acid and NF-κB signalling pathways regulate only osteoblast dedifferentiation, but not cell shape changes and migration, while the complement and actomyosin systems are required for osteoblast cell shape changes and migration, but not for dedifferentiation.”